# Lack of Hikeshi activates HSF1 activity under normal conditions and disturbs the heat-shock response

Shingo Kose[1], Kenichiro Imai[2], Ai Watanabe[1], Akira Nakai[3], Yutaka Suzuki[4], Naoko Imamoto[1]

Hikeshi mediates the nuclear import of the molecular chaperone HSP70 under heat-shock (acute heat stress) conditions, which is crucial for recovery from cellular damage. The cytoplasmic function of HSP70 is well studied, but its nuclear roles, particularly under nonstressed conditions, remain obscure. Here, we show that Hikeshi regulates the nucleocytoplasmic distribution of HSP70 not only under heat-shock conditions but also under nonstressed conditions. Nuclear HSP70 affects the transcriptional activity of HSF1 and nuclear proteostasis under nonstressed conditions. Depletion of Hikeshi induces a reduction in nuclear HSP70 and up-regulation of the mRNA expression of genes regulated by HSF1 under nonstressed conditions. In addition, the heat-shock response is impaired in Hikeshi-knockout cells. Our results suggest that HSF1 transcriptional activity is tightly regulated by nuclear HSP70 because nuclear-localized Hsp70 effectively suppresses transcriptional activity in a dose-dependent manner. Furthermore, the cytotoxicity of nuclear pathologic polyglutamine proteins was increased by Hikeshi depletion. Thus, proper nucleocytoplasmic distribution of HSP70, mediated by Hikeshi, is required for nuclear proteostasis and adaptive response to heat shock.

## Introduction

Proper nucleocytoplasmic translocation of functional proteins through the nuclear pore complex is an essential cellular process for eukaryotic cells that affects chromatin organization, gene expression, signal transduction, and protein homeostasis (proteostasis). In general, evolutionarily conserved members of the importin-$\beta$ family members function as nucleocytoplasmic transport receptors, which shuttle continuously between the nucleus and cytoplasm through the nuclear pore complex and mediate the localization-signal–dependent translocation of their cargoes either into or out of the nucleus (reviewed in Wente and Rout [2010] and Paci et al [2021]). Dysregulation of the nucleocytoplasmic transport system induces aberrant intracellular localization of various functional proteins, causing various human diseases, including neurodegenerative disorders (reviewed in Mor et al [2014], Bitetto and Di Fonzo [2020], and Hutten and Dormann [2020]).

Hikeshi was originally identified as a nuclear import receptor that mediates the nuclear import of cytosolic molecular chaperone HSP70s (HSPA1A/HSPA1B/HSPA8 in humans) under heat-shock (acute heat stress) conditions (Kose et al, 2012). Cellular stresses affect not only the structural stability of folded proteins but also the nucleocytoplasmic transport system. In response to various cellular stresses, such as heat, oxidative, and UV stresses, conventional importin-$\beta$–mediated nucleocytoplasmic transport is down-regulated (Furuta et al, 2004; Miyamoto et al, 2004; Kelley & Paschal, 2007; Kodiha et al, 2008; Yoshimura et al, 2013; Ogawa & Imamoto, 2018). However, HSP70 is efficiently translocated from the cytoplasm into the nucleus in response to heat shock (Pelham, 1984; Velazquez & Lindquist, 1984; Welch & Feramisco, 1984; Ogawa & Imamoto, 2018). This heat-shock–responsive (HSR) nuclear import of HSP70 is mediated by Hikeshi, which does not belong to the importin-$\beta$ family. Hikeshi mediates the nuclear import of HSP70 in *Schizosaccharomyces pombe* (Oda et al, 2014) and *Arabidopsis* (Koizumi et al, 2014). Currently, in humans, Hikeshi is the only carrier molecule of HSP70s for nuclear import. Dysfunction of Hikeshi inhibits the nuclear import of HSP70s under heat-shock conditions and then impairs recovery of the cell from stress damage, leading to cell death or aberrant cell growth (Kose et al, 2012; Rahman et al, 2017; Yanoma et al, 2017). Exogenously expressed classical NLS-tagged HSP70, which is carried into the nucleus by the importin-$\alpha/\beta$ pathway, significantly suppresses cell death of Hikeshi-depleted HeLa cells caused by heat shock (Kose et al, 2012). These results indicate that the nuclear function of HSP70 is crucial to revert the phenotypes caused by Hikeshi depletion.

The heat-shock response is an adaptive mechanism by which cells counter proteotoxic stresses that induce heat-shock proteins (HSPs) (reviewed in Akerfelt et al [2010]). HSPs function as molecular chaperones with their regulator cochaperones (reviewed in Kim et al [2013]). HSPs facilitate refolding of unfolded proteins and restore proteostasis. Heat-shock factor 1 (HSF1) is a master

[1]Cellular Dynamics Laboratory, RIKEN Cluster for Pioneering Research, Wako, Japan   [2]Cellular and Molecular Biotechnology Research Institute, National Institute of Advanced Industrial Science and Technology (AIST), Tokyo, Japan   [3]Department of Biochemistry and Molecular Biology, Yamaguchi University School of Medicine, Ube, Japan   [4]Department of Computational Biology and Medical Sciences, Graduate School of Frontier Sciences, The University of Tokyo, Kashiwa, Japan

Correspondence: nimamoto@riken.jp; skose@riken.jp

transcription factor of the heat-shock response and regulates the mRNA expression levels of heat-shock–responsive genes involving HSPs and cochaperones. HSF1 is activated by multistep processes (reviewed in Anckar and Sistonen [2011] and Gomez-Pastor et al [2018]). In response to heat shock, HSF1 is converted to a transcriptionally active homotrimer from an inactive monomer that binds to a consensus heat-shock element (HSE) of the promoter regions of HSR genes (Xiao et al, 1991; Sarge et al, 1993), and its transcriptional activity of HSF1 is regulated by various post-translational modifications, including phosphorylation (Guettouche et al, 2005; Zheng et al, 2016; reviewed in Gomez-Pastor et al [2018]).

HSP70 is an evolutionarily conserved protein functioning as a molecular chaperone that plays critical roles in diverse cellular activities, such as nascent protein folding, prevention of protein aggregation, refolding of misfolded protein, or protein degradation (reviewed in Rosenzweig et al [2019]). In humans, constitutive Hsc70 (HSPA8) and two inducible Hsp70 isoforms (HSPA1A/HSPA1B) are major cytosolic HSP70 proteins. The chaperone activity of HSP70 is coupled with the ATPase cycle of HSP70, which is mainly regulated by two groups of cochaperones, J domain proteins (JDPs) (Hsp40 family, DNAJA/DNAJB/DNAJC) (reviewed in Kampinga and Craig [2010]) and nucleotide exchange factors (Hsp110 and BAG families) (reviewed in Bracher and Verghese [2015]). JDPs bind to HSP70 and stimulate the ATPase activity of HSP70, facilitating tight binding of HSP70 to substrates. Nucleotide exchange factors accelerate the nucleotide exchange of ADP to ATP bound to HSP70, which facilitates the release of substrates from HSP70, leading to refolding of the substrates. The number of HSP genes tends to increase in proportion to increasing genes in the genome, evolving diverse proteostasis networks (Powers & Balch, 2013). However, the specific substrates and cellular functions of the HSP70 network, coupled with diverse cochaperones, have largely remained elusive. Furthermore, although the cytoplasmic functions of HSP70 in proteostasis have been elucidated extensively, relatively little is known about the nuclear functions of HSP70.

Hikeshi is a nonessential gene in human and mouse cultured cells and yeast. However, recent genetic studies have shown that Hikeshi has an important role in the maintenance of organismal physiology. Hikeshi-KO mice die soon after birth (Fernández-Valdivia et al, 2006; our unpublished results). Furthermore, in humans, the homozygous p.Val54Leu or p.Cys4Ser mutation in the *Hikeshi/C11orf73* gene causes the expression of unstable Hikeshi proteins, leading to hypomyelinating leukoencephalopathies (Edvardson et al, 2016; Vasilescu et al, 2017). These reports suggested that Hikeshi has ambiguous functions under physiological conditions. At the cellular level, Hikeshi interacts with HSP70s (HSPA1A/HSPA1B/HSPA8) and shuttles between the nucleus and cytoplasm. Although the activity of Hikeshi is presumed to be associated with nuclear HSP70 functions, we do not know whether Hikeshi has cellular roles under nonstressed conditions. Furthermore, the function of nuclear HSP70, particularly under nonstressed conditions, remains largely unknown.

Orthologs of the importin-β family molecule nucleocytoplasmic transport receptors were established early in the emergence of eukaryotes and likely before the last eukaryotic common ancestor (O'Reilly et al, 2011). The search for evolutionary lineages of Hikeshi showed that Hikeshi orthologs were widely distributed in eukaryotic organisms but not in prokaryotes (Fig S1). These results indicate that Hikeshi, like the importin-β family, is present in most eukaryotic cells that evolved soon after the last eukaryotic common ancestor. It must be noted that HSP70 is one of the most evolutionarily preserved proteins in both prokaryotes and eukaryotes. The evidence that Hikeshi was acquired only after the emergence of eukaryotic cells implicates that the function of Hikeshi is connected to the cell nucleus.

In this study, we showed that Hikeshi regulates the nuclear distribution of HSP70 even in the absence of heat shock, affecting the transcriptional activity of HSF1 and nuclear proteostasis under nonstressed conditions. In Hikeshi-KO cells, many HSF1-regulated genes were selectively up-regulated under nonstressed conditions. The absence of nuclear HSP70 caused nuclear protein instability and higher toxicity of nuclear pathologic polyglutamine proteins. Furthermore, the heat-shock response was impaired in Hikeshi-KO cells.

# Results

## KO of the Hikeshi gene induces a reduction in nuclear HSP70 under nonstressed conditions

We initially showed that Hikeshi is required for cell survival after heat shock (acute heat stress) (Kose et al, 2012), but our subsequent studies showed that the effect of Hikeshi is more complicated (Rahman et al, 2017). Depletion of Hikeshi reduced the survival of HeLa cells but increased the survival of hTERT-RPE1 cells in response to proteotoxic stress. We generated Hikeshi-KO cells using CRISPR–Cas9–mediated gene editing (Rahman et al, 2017) and then analyzed how the cellular localization of HSP70 was affected by Hikeshi-KO. In accordance with previous reports (Kose et al, 2012; Rahman et al, 2017), heat-shock–induced nuclear import of HSP70 was strongly inhibited in Hikeshi-KO cells (Fig 1A). The addition of MG132, a proteasome inhibitor well known to induce cellular proteotoxic stress, also induced nuclear accumulation of HSP70 but to lesser extent than those of heat shock. The addition of MG132 also induced significant nucleolar accumulation of HSP70, which was also seen in cells under heat-shock conditions. This MG132-induced nuclear/nucleolar accumulation of HSP70 was strongly inhibited in the Hikeshi-KO cells (Fig 1B). The effect of Hikeshi-KO on heat-shock– and MG132-induced nuclear/nucleolar accumulation of HSP70 was similarly seen in HeLa cells and hTERT-RPE1 cells (see Rahman et al [2017] for under heat-shock conditions). These results showed that Hikeshi mediated the nuclear import of HSP70 under proteotoxic stress conditions.

Under nonstressed conditions, HSP70 was largely localized in the cytoplasm, but a small portion of HSP70 resided in the nucleus in both HeLa and hTERT-RPE1 cells (Fig 1A and C). It must be noted that HSP70 is known to be one of the most abundant proteins. A previous report showed that in HeLa cells, HSPA8 and HSPA1A/HSPA1B were calculated as $9.5 \times 10^6$ and $5.6 \times 10^6$ copies per cell, respectively, whereas the median abundance of protein was $2.1 \times 10^4$ copies per cell, and 92.6% of proteins identified were expressed in a range from 100 to $10^6$ copies per cell (Kulak et al, 2014). Therefore, even if 1% of HSP70 resided in the nucleus, nuclear HSP70 was estimated to be $7.6 \times 10^4$ copies per cell, which is more than threefold higher than the median abundance of protein expressed in HeLa cells. These

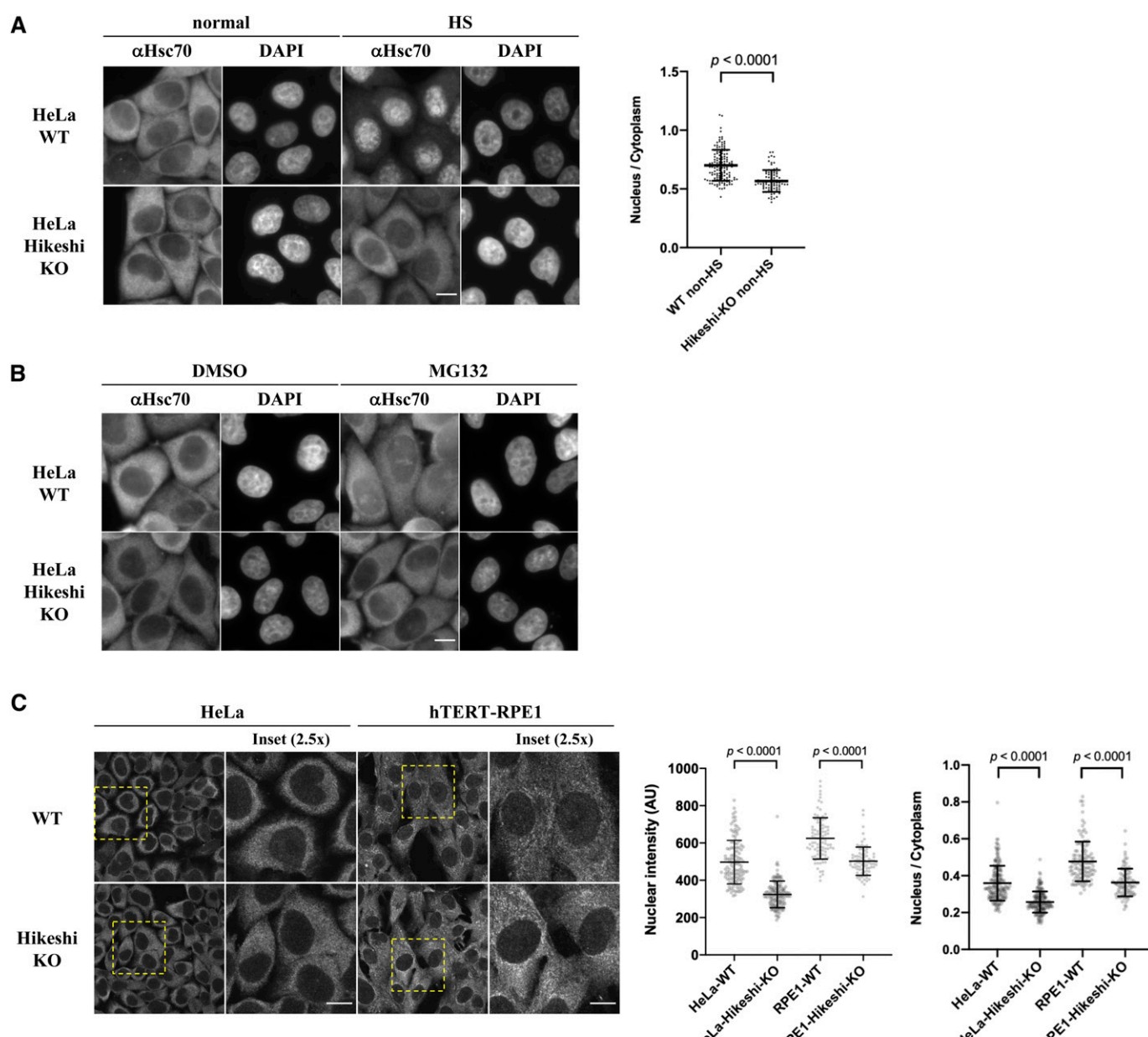

**Figure 1. Hikeshi is required for the nuclear import of HSP70 under both nonstressed and proteotoxic stress conditions.**
**(A)** Hikeshi mediates the nuclear import of HSP70 under heat-shock conditions. WT or Hikeshi-KO HeLa cells were incubated for 1 h at 43°C (HS) or 37°C (normal). **(B)** Hikeshi mediates the nuclear import of HSP70 under treatment with the proteasome inhibitor MG132. Cells were treated with 10 $\mu$M MG132 or DMSO for 6 h. In (A, B), DNA was counterstained with DAPI. **(C)** Depletion of Hikeshi affects the nuclear localization of HSP70 under nonstressed conditions. In (A, B, C), cells were fixed and immunostained with anti-Hsc70 antibodies (1B5). **(A, B, C)** Images were captured with an Olympus BX51 microscope (A, B) or FV1200 confocal microscope (C). Scale bars show 10 $\mu$m. In (A, C), nuclear and cytoplasmic intensities were measured with ImageJ software. Nuclear intensities (left graph in C) and nuclear intensities relative to cytoplasmic intensities (A and right graph in C) of each cell were plotted. Error bars indicate means ± SD. Statistical analyses were performed using Welch's two-sided $t$ tests.

levels of nuclear HSP70 were detected by immunofluorescence analysis. The quantification of the immunofluorescence images showed that nuclear HSP70 under nonstressed conditions was explicitly lower in both HeLa and hTERT-RPE1 Hikeshi-KO cells compared with WT cells (Fig 1C, also see Fig 1A), showing that Hikeshi affected the subcellular localization of HSP70 under nonstressed conditions, probably by mediating its nuclear import.

## KO of the Hikeshi gene induces up-regulated expression of HSF1-regulated genes under nonstressed conditions

HSP70 plays important roles in cellular proteostasis. However, it is unclear whether the reduction in nuclear HSP70 proteins affects cellular activities under nonstressed conditions. To obtain information, we performed RNA sequencing (RNA-seq) analyses of HeLa

WT and Hikeshi-KO cells. We found that the mRNA expression of 140 or 155 genes was up-regulated (fold change > 1.5) (Fig 2A) or down-regulated (fold change < 0.7) (Fig S2A) in Hikeshi-KO cells compared with WT cells under nonstressed conditions, respectively. The lists of up-regulated and down-regulated genes are shown in Table S1. Gene ontology (GO) term analysis of up-regulated genes showed strong enrichment for GO terms related to protein folding (Fig 2B), and down-regulated genes showed enrichment for GO terms related to negative regulation of lipase activity (Fig S2B). Up-regulated genes related to protein folding include the HSP70 chaperone family (*HSPA1A*, *HSPA1B*, *HSPA1L*, and *HSPA6*) and its regulators (*DNAJA4*, *DNAJB1*, *HSPH1*, and *BAG3*) (Fig 2B and C, left panel). Notably, these eight genes and *DEDD2* gene, which is listed as up-regulated genes in Table S1, have been reported to be regulated by the transcription factor HSF1 (Page et al, 2006; Vilaboa et al, 2017; Kovács et al, 2019). Therefore, we focused on these nine genes and validated RNA-seq results using quantitative RT-PCR (qPCR). qPCR results confirmed that these nine genes were significantly up-regulated in Hikeshi-KO cells under nonstressed conditions (Fig 2C, light panel). siRNA-mediated depletion of Hikeshi, by which protein expression levels of Hikeshi were reduced to ~10%, also showed that most of these nine genes were up-regulated under nonstressed conditions, with some exceptions (Fig S2C). We found some variability in up-regulation levels of these nine genes resulting from Hikeshi-KO cells and siHikeshi-treated cells. Unlike RNA-seq results with Hikeshi-KO cells (Fig 2C), in siHikeshi-treated cells, *HSPA1L* was not up-regulated in RNA-seq experiment, *BAG3* was not significantly up-regulated in both RNA-seq and qPCR experiments, and *DEDD2* was not up-regulated in qPCR experiment (Fig S2C). We presume these discrepancies are because of protein levels of Hikeshi remaining in siHikeshi-treated cells. Next, to confirm that mRNA expressions of these nine genes are regulated by HSF1, we treated HeLa WT and Hikeshi-KO cells with siRNA targeting *HSF1* and examined mRNA expression levels with qPCR. Consistent with previous reports (Page et al, 2006; Vilaboa et al, 2017; Kovács et al, 2019), siRNA-mediated depletion of HSF1 induced down-regulation of mRNA expression of these genes under nonstressed conditions in HeLa WT and Hikeshi-KO cells (Fig 2D), indicating that mRNA expression of these genes was regulated by HSF1.

Similarly, RNA-seq data of hTERT-RPE1 WT and Hikeshi-KO cells showed that 272 genes were up-regulated in Hikeshi-KO cells under nonstressed conditions compared with WT cells (fold change > 1.5) (Fig S3A and Table S1). GO term analysis showed that these up-regulated genes in the Hikeshi-KO cells were enriched with protein-folding genes that included the HSP70 chaperone family (*HSPA1A* and *HSPA1B*) and its regulators (*DNAJB4*, *HSPA4L*, *HSPH1*, and *BAG3*) (Fig S3B). RNA-seq results showed that mRNA expressions of HSF1-regulated genes, listed in Fig 2C, were significantly up-regulated in the Hikeshi-KO hTERT-RPE1 cells under nonstressed conditions, although mRNA expressions of *HSPA6*, *HSPA1L*, and *DNAJA4* could not be confirmed because of their low expression levels (Fig S3C, left panel). qPCR results for validation of RNA-seq also showed that *HSPA1L* was not significantly up-regulated and that *DNAJA4* could not be detected probably because of their low expression levels (Fig S3C, right panel).

Taken together, these data show that under nonstressed conditions, depletion of Hikeshi affects the expression of various genes

and particularly induces up-regulation of mRNA expression of genes regulated by HSF1.

## Nuclear HSP70 regulates the transcriptional activity of HSF1

As shown in Fig 1C, nuclear HSP70 was explicitly lower in Hikeshi-KO cells under nonstressed conditions. Simultaneously, HSF1-regulated genes were up-regulated. We wondered whether these two were directly connected. Although the binding of HSP70 to HSF1 is considered to negatively regulate the transcriptional activity of HSF1 during recovery from stress (Abravaya et al, 1992; Baler et al, 1992; Shi et al, 1998; Zheng et al, 2016; Krakowiak et al, 2018; Masser et al, 2019; Peffer et al, 2019; Kmiecik et al, 2020), it is not clear whether nuclear-localized HSP70 is responsible for regulating the transcriptional activity of HSF1 under nonstressed conditions. To address this question, we constructed HeLa cells stably expressing EGFP-NLS–tagged Hsc70 (HSPA8) and examined the mRNA expression levels of HSF1-regulated genes in these cells. In the established stable cell lines, EGFP-NLS-Hsc70 proteins were properly localized in the nucleus (Fig 3A), and their expression levels were much lower than those of endogenous HSP70s (Fig 3B). The RNA-seq analysis performed in these stably transformed cells under nonstressed conditions showed that all HSF1-regulated genes listed in Fig 2C were down-regulated in HeLa cells stably expressing NLS-Hsc70 (Fig 3C). qPCR analyses showed that mRNA expression levels of these genes were down-regulated in two stable clonal cells however that *HSPA1L*, *DNAJA4*, and *BAG3* were only weakly down-regulated in one of the two stable clonal cells (Fig S4A). Transient expressions of NLS-Hsc70, whose expression levels were higher than stable clones, induced down-regulation of these genes, except for *HSPA1L* (Fig S4B). mRNA expression levels of *HSPA1L* is high in the testis but very low in most tissue (Hageman & Kampinga, 2009). Consistently, our RNA-seq data showed that mRNA expression levels of *HSPA1L* is very low under nonstressed conditions in HeLa and hTERT-PRE1 cells (see Fig S5A for HeLa cells). This may make it difficult to assess suppression of *HSPA1L* expression with qPCR technique. Overall, we conclude that results of qPCR basically support our RNA-seq data, although there are differences for genes whose expression levels are low.

To confirm the inhibitory effect of nuclear HSP70 on the transcriptional activity of HSF1, we performed a firefly luciferase (Fluc) reporter assay driven by HSEs. HeLa WT and Hikeshi-KO cells were transfected with the HSE-Fluc reporter plasmid, and then luciferase activities were measured using these cell lysates. As shown in Fig 3D, the reporter luciferase activities were up-regulated 3.6-fold in the Hikeshi-KO cells relative to WT cells under nonstressed conditions. The increased luciferase activities in the Hikeshi-KO cells were strongly suppressed by cotransfection with the NLS-Hsc70–expressing plasmid in a dose-dependent manner (Fig 3E). Cotransfection of nontagged or nuclear export signal-tagged Hsc70-expressing plasmid did not suppress but rather increased the luciferase activity (Fig S4C). The enhanced effect on luciferase activities of nontagged or NES-tagged Hsc70 may indicate that cytoplasmic luciferases that expressed through the HSE promoter are stabilized by chaperone function of cytoplasmic Hsc70. These results show that HSP70 that localized in the nucleus inhibited transcriptional activation through the HSE promoter. Consistently,

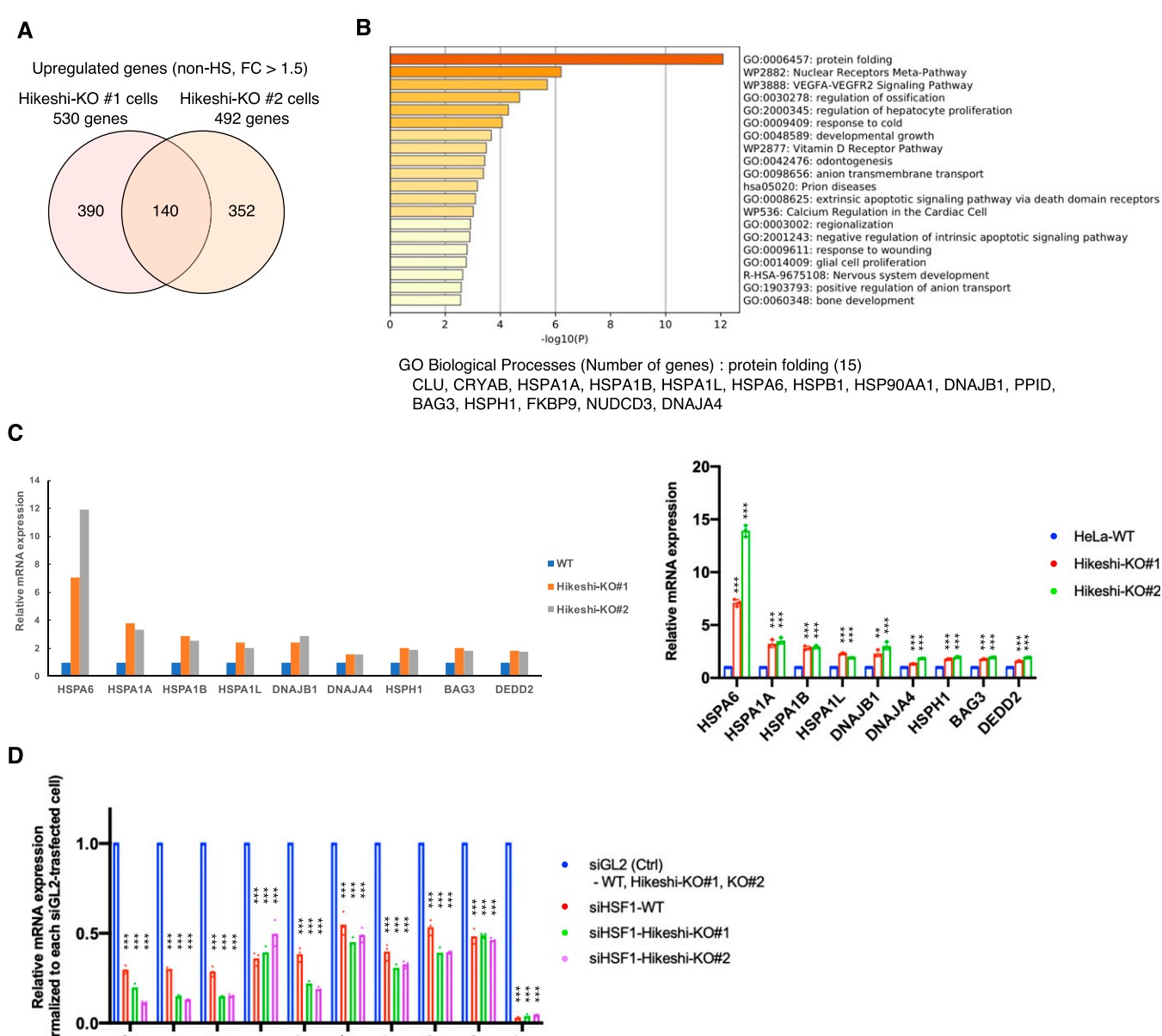

**Figure 2. HSF1-regulated genes are up-regulated under nonstressed conditions in Hikeshi-KO cells.**
**(A)** Venn diagram showing the number of genes up-regulated under nonstressed conditions in Hikeshi-KO HeLa cells (clones #1 and #2) compared with WT cells (fold change > 1.5). mRNA expression levels were quantified by RNA-seq (single replicate for each cell line). **(B)** Gene ontology enrichment analysis of 140 genes commonly up-regulated in the two different Hikeshi-KO HeLa clones by Metascape. **(C)** mRNA expression levels of HSF1-regulated genes under nonstressed conditions quantified by RNA-seq (n = 1, left panel) and qPCR (n = 3 biologically independent experiments, right panel). mRNA expression levels of HSF1-regulated genes in two HeLa Hikeshi-KO cells (clones #1 and #2) relative to that of WT cells are shown. **(D)** mRNA expression levels of HSF1-regulated genes under nonstressed conditions in siHSF1-treated cells. HeLa WT and Hikeshi-KO cells were transfected with siRNA targeting GL2 (siGL2, control) or HSF1 (siHSF1). mRNA expression levels were quantified by qPCR (n = 3 biologically independent experiments). In each cell, mRNA expression levels in siHSF1-transfected cells relative to that in siGL2-transfected cells were shown. Error bars indicate ± SD. Statistical significance was determined using unpaired *t* test (NS, no significance, *P* > 0.05; ***P* < 0.001, ***P* < 0.01).

cotransfection with the Hikeshi-expressing plasmid suppressed the luciferase activities in the Hikeshi-KO cells (Fig 3F). These results suggest that nuclear-localized HSP70, mediated by Hikeshi, negatively regulates the transcriptional activity of HSF1 that was activated in Hikeshi-KO cells under nonstressed conditions.

Our results suggest that Hikeshi mediates the nuclear import of HSP70 under nonstressed conditions in addition to heat-shock conditions and that nuclear-localized HSP70 under nonstressed conditions has an important role in the negative regulation of HSF1 transcriptional activity. Therefore, lack of Hikeshi, which leads to a decrease in nuclear HSP70, causes dysregulation of HSF1 transcriptional activity.

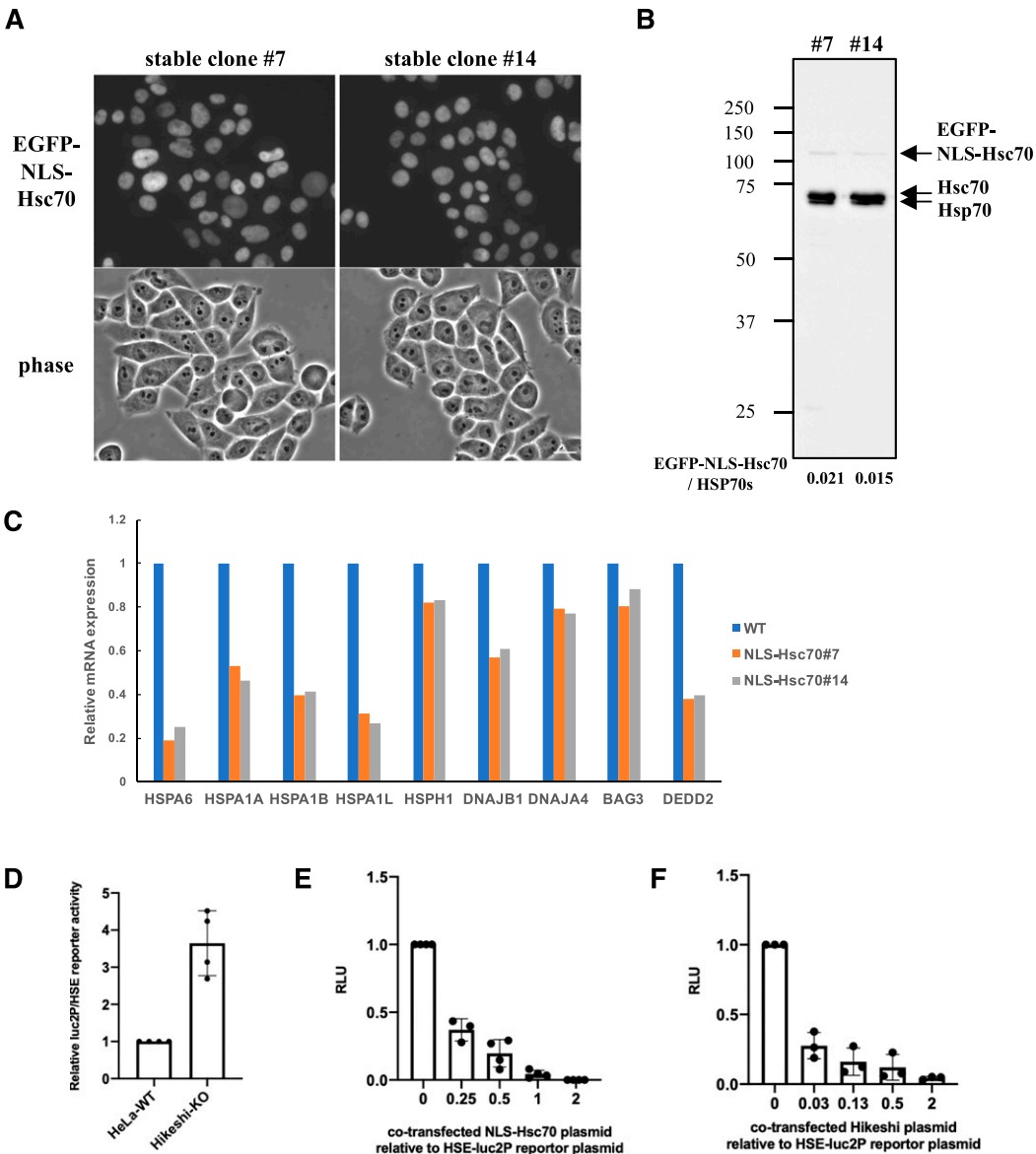

**Figure 3. Nuclear HSP70 suppresses the transcriptional activity of HSF1.**
**(A)** Cellular localization of EGFP-NLS-Hsc70 in HeLa cells stably expressing EGFP-NLS-Hsc70 (clones #7 and #14). **(B)** EGFP-NLS-Hsc70 protein stably expressed in the HeLa cell lines (clones #7 and #14) and endogenous Hsc70 (HSPA8) and Hsp70 (HSPA1) were detected by immunoblotting with anti-Hsc70/Hsp70 antibodies (1H5-1). Band intensities were measured using ImageJ software. Ratios of EGFP-NLS-Hsc70 to endogenous Hsc70 and Hsp70 proteins are shown. **(C)** mRNA expression levels of 9 HSF1-regulated genes, selected in Fig 2C, under nonstressed conditions quantified by RNA-seq (single replicate for each cell line). mRNA expression levels of HSF1-regulated genes in HeLa cells stably expressing EGFP-NLS-Hsc70 (clones #7 and #14) relative to that in HeLa WT cells are shown. **(D)** Fluc reporter gene expression driven by HSEs is highly activated in Hikeshi-KO cells. pGL4.41[luc2P/HSE/Hygro] vectors were cotransfected with pNL1.1TK (Nluc/TK) vectors, which were used as transfection controls, into WT and Hikeshi-KO HeLa cells. At 1 d posttransfection, Fluc and Nluc activities were measured using a Nano-Glo Dual-Luciferase Reporter Assay (Promage). **(E)** NLS-Hsc70 suppresses Fluc reporter gene expression driven by the HSE promoter. Hikeshi-KO cells were cotransfected with 37 ng of pGL4.41[luc2P/HSE/Hygro] vector, and the plasmid expressing EGFP-NLS-Hsc70 at ratios shown in the figure. **(F)** Exogenously expressed Hikeshi in Hikeshi-KO cells suppresses Fluc reporter gene expression driven by the HSE promoter. Hikeshi-KO cells were cotransfected with 37 ng of pGL4.41[luc2P/HSE/Hygro] vector, and the plasmid expressing FLAG-Hikeshi at ratios shown in the figure. In (E, F), Fluc activities at 1 d posttransfection were measured using a luciferase assay system (Promega). Error bars indicate ± SD.

## Hikeshi-mediated nuclear-localized HSP70 functions in nuclear protein stability

As shown in Fig 1C, nuclear HSP70 under nonstressed conditions was explicitly lower in Hikeshi-KO cells compared with WT cells. Because HSP70 plays essential roles in proteostasis, conformational instability

of nuclear proteins may be facilitated in Hikeshi-KO cells, leading to dysfunction of nuclear proteins. To test this possibility, we used the NLS-tagged Fluc (NLS-Fluc) protein and measured its luciferase activity as an indicator of nuclear protein stability. Fluc is a well-characterized substrate of HSP70 that is frequently used to monitor the chaperone capacity of HSP70 (Schröder et al, 1993; Frydman et al, 1994; Terada et al,

1997). Cells were transfected with a plasmid expressing NLS-Fluc and then treated with the protein synthesis inhibitor cycloheximide (CHX) for 2 h to exclude the luciferase activity of newly synthesized NLS-Fluc proteins. The luciferase activities of NLS-Fluc in cells untreated with CHX were stipulated to be 100% (see Fig 4A). After treatment with CHX, NLS-Fluc expressed in nuclei of WT cells retained ~43% of the luciferase activity, whereas NLS-Fluc in the Hikeshi-KO cells retained ~36% of the luciferase activity, showing that nuclear Fluc proteins in the Hikeshi-KO cells were less stable than those in WT cells (Fig 4B). Notably, these effects were more significant with the NLS-Fluc-R188Q single mutant (SM), which tends to be more unstable than wild-type Fluc (Gupta et al, 2011). Although luciferase activity is influenced by both protein stability and refolding activity of chaperones, our data showing that luciferase activity of nuclear-localized Fluc-SM was more influenced than that of nuclear-localized Fluc-WT in the Hikeshi-KO cells may suggest that the function of nuclear HSP70 confers with protein stability. NLS-Fluc-SM in WT and Hikeshi-KO cells retained 34% and 23% of luciferase activity, respectively, within 2 h of treatment with CHX (Fig 4B). Coexpression of NLS-tagged HSP70 in both WT and Hikeshi-KO cells alleviated the luciferase activity of nuclear Fluc proteins in a dose-dependent manner (Fig 4C). Next, we examined the protein stability of nontagged Fluc proteins, which are localized in the cytoplasm. As shown in Fig 4D, there were no differences in the luciferase activity of cytoplasmic Fluc, either wild type or SM mutant, between WT and Hikeshi-KO cells. These results showed that Hikeshi

affects protein stability in the nucleus but not in the cytoplasm under nonstressed conditions. Altogether, nuclear-localized HSP70, which is a function of Hikeshi, possesses an important function in nuclear proteostasis under nonstressed conditions.

## Hikeshi reduces cellular apoptosis induced by nuclear polyQ expression

Expansion of polyglutamine (polyQ) repeats leads to protein misfolding and aggregation, causing inherited neurodegenerative diseases. HSPs and their cochaperones are known to act as molecular chaperones to suppress these aggregations and cell death. To better understand the cellular functions of Hikeshi and nuclear HSP70s, we examined the impact of Hikeshi dysfunction on polyQ aggregate formation and cell viability. As previously reported (Onodera et al, 1997), transiently expressed pathologic polyQ81 from human DRPLA fused to EGFP (81Q-EGFP) localized both in the nucleus and cytoplasm and formed perinuclear aggregates in HeLa cells (data not shown). Therefore, to detect compartment-specific cytotoxic effects (i.e., effect of nuclear polyQ aggregates or cytoplasmic polyQ aggregates), we transiently expressed NLS- or NES-tagged Q81-EGFP proteins that exclusively localize in the nucleus or cytoplasm, respectively, in WT or Hikeshi-KO HeLa cells. PolyQ81-NLS-EGFP and polyQ81-NES-EGFP proteins were properly localized in the nucleus and cytoplasm, respectively (Fig 5A). Next, we

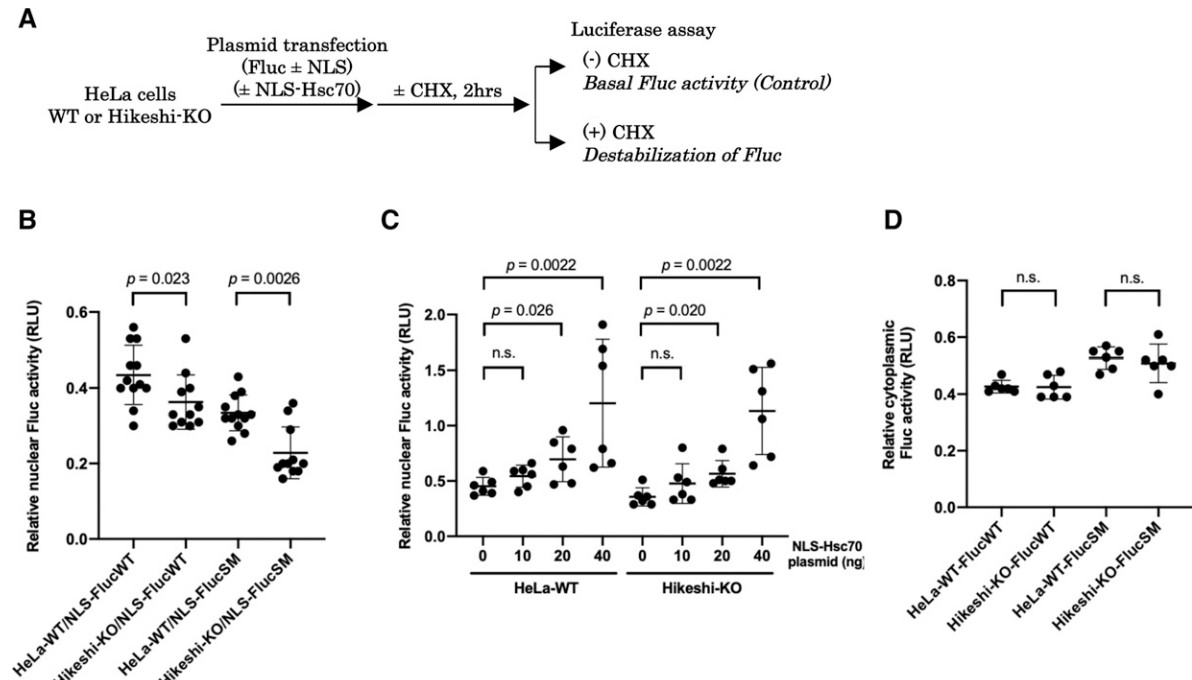

**Figure 4.  Nuclear HSP70 functions in the protein activity of nuclear luciferase monitoring proteins.**
**(A)** Schema of experimental procedures. **(B, C, D)** In (B, C, D), at 1 d posttransfection, cells were incubated with or without 25 µg/ml of the protein synthesis inhibitor cycloheximide (CHX) for 2 h, and then Fluc activities were measured using a luciferase assay system (Promega). Ratios of Fluc activities of CHX-treated cells to that of untreated cells are plotted in the graph. Error bars indicate means ± SD. Statistical analyses were performed using Mann–Whitney two-tailed analysis (n.s., no significance, $P > 0.05$). **(B)** Protein stability of nuclear Fluc decreases in Hikeshi-KO cells. EGFP-NLS–tagged Fluc-WT or SM (more unstable Fluc mutant) was transiently expressed in HeLa WT or Hikeshi-KO cells. **(C)** Nuclear HSP70 suppresses the decrease in nuclear Fluc protein stability. HeLa WT and Hikeshi-KO cells were cotransfected with 50 or 60 ng of EGFP-NLS-Fluc-WT-expressing plasmid and NLS-Hsc70–expressing plasmid. The amount of transfected plasmid expressing NLS-Hsc70 is shown in the figure. **(D)** Protein stability of cytoplasmic Fluc-WT and SM. EGFP-Fluc-WT or SM was transiently expressed in HeLa WT or Hikeshi-KO cells.

measured the effect of pathologic polyQ proteins on apoptosis by using the caspase-Glo 3/7 assay system. As shown in Fig 5B, expression of NLS-polyQ81, but not NES-polyQ81, significantly induced up-regulation of caspase 3/7 activity, suggesting that nuclear expression of the pathologic polyQ protein is highly toxic to the cells. Furthermore, the cytotoxicity of nuclear polyQ81 was higher in Hikeshi-KO cells compared with WT cells (Fig 5B). Coexpression of Hikeshi (Fig 5C) or NLS-Hsc70 (Fig 5D) in the Hikeshi-KO cells significantly reduced the apoptosis induction caused by nuclear polyQ81 proteins. These results suggest that the cellular activity of Hikeshi contributes to relieving the cytotoxicity of nuclear pathologic polyQ through function of nuclear HSP70s.

### Heat-shock response is impaired in the Hikeshi-KO cells

In Hikeshi-KO cells, the expression of HSF1-regulated genes was up-regulated under nonstressed conditions (Fig 2C). We previously showed that the activation of HSF1, characterized by phosphorylation and nuclear stress granule (nSG) formation, was sustained for a longer time after recovery from heat shock in Hikeshi-knockdown and KO cells (Kose et al, 2012; Rahman et al, 2017). These results may indicate that activation and attenuation of HSF1 in response to heat shock was impaired in Hikeshi-KO cells. To investigate a possible impairment of the heat-shock response, WT and Hikeshi-KO cells were exposed to heat shock (HS) for 1 h and were returned to normal temperature for 1.5 or 3 h (R1.5h or R3h, respectively), and then, we performed RNA-seq analyses at each time point. In WT HeLa cells treated with heat shock and without any recovery, mRNA expression of 162 genes was induced more than twofold under heat-shock conditions relative to nonstressed conditions. We categorized these 162 genes as HSR genes.

RNA-seq analyses showed that HSR mRNA expression of HSR genes in Hikeshi-KO cells was reduced compared with that in WT cells. As shown in Fig 6A left panels, medians of mRNA expression levels of HS relative to non-HS in two Hikeshi-KO cells are 2.28 and 2.27, whereas median of mRNA expression levels of HS relative to non-HS of WT cells is 2.91. To make easier to see, we also plotted mRNA expression levels of HSR genes at each time point in Hikeshi-KO cells relative to WT cells (Fig 6A, right panels). As shown in the both panels, relative mRNA expression levels at HS in the Hikeshi-KO cells were reduced compared with that in WT cells. After HS and beyond, as shown in Fig 6A left panels, mRNA expressions of HSR genes in WT cells peaked at R1.5h and then decreased at R3h (the median of mRNA expression levels of HS, R1.5h, and R3h relative to non-HS is 2.91, 3.53, and 2.36, respectively). On the other hand, mRNA expressions of HSR genes in two Hikeshi-KO cells gradually increased from non-HS to R3h (the median of mRNA expression levels of HS, R1.5h, and R3h relative to non-HS is 2.28 and 2.27, 3.06 and 2.92, and 3.54 and 3.71, respectively) (Fig 6A, left panel). As a result, mRNA expression levels of HSR genes at R3h were higher in Hikeshi-KO cells than in WT cells (Fig 6A, right panel). These results suggest that in the Hikeshi-KO cells, up-regulation of HSR genes in response to heat shock is weakened and is sustained at HS and beyond (up to 3 h after recovery to normal temperature after HS).

Next, we analyzed the mRNA expression of the HSF1-regulated genes listed in Fig 2C. In Fig 6B and C, like HSR genes described

above, mRNA expression levels of HS (and beyond) relative to non-HS in WT cells or two Hikeshi-KO cells are shown in left panels, and mRNA expression levels of two Hikeshi-KO cells relative to WT cells at each time points are shown in right panels. RNA-seq analysis (Fig 6B) and qPCR analysis (Fig 6C) both showed that HSR up-regulation of these HSF1-regulated genes was weakened in Hikeshi-KO cells and sustained during recovery periods like HSR genes. At R4.5h, mRNA expression levels of most of these HSF1-regulated genes were higher in Hikeshi-KO cells than that in WT cells (Fig 6C, right panel).

Furthermore, RNA-seq data showed that more than half of the HSP family genes (HSPA, HSPH, DNAJA, and DNAJB), including some of HSF1-regulated genes, were up-regulated under nonstressed conditions in Hikeshi-KO cells (the median of mRNA expression levels in Hikeshi-KO#1 and KO#2 cells relative to that in WT cells is 1.31 and 1.06, respectively) (Fig S7). As in the case for the HSF1-regulated genes (Fig 6B and C), the expression levels of most HSP genes at R1.5h and R3h were lower in the Hikeshi-KO cells than in the WT cells (Fig S7). Taken together, our results showed that depletion of Hikeshi impairs the appropriate heat-shock response in a number of ways.

## Discussion

Proper regulation of the subcellular localization of functional proteins is essential for eukaryotes. In this study, we discovered that Hikeshi, which was acquired only after eukaryotes evolved, regulates the nuclear localization of HSP70 under nonstressed conditions. HSP70 is a well-known molecular chaperone known to possess essential roles in proteostasis in the cytoplasm under nonstressed and stress conditions. On the other hand, its nuclear function, particularly under nonstressed conditions, is less characterized. Here, we showed that loss of Hikeshi, which induces decrease of nuclear HSP70, affects the expression of diverse genes, including HSF1-regulated transcription and nuclear proteostasis, under nonstressed conditions. Furthermore, the heat-shock response is impaired in Hikeshi-KO cells. Our results suggest that nuclear HSP70 is strongly associated with the adaptive response of HSF1 transcriptional activity.

HSP70 is mainly localized in the cytoplasm under nonstressed conditions, but it accumulates in the nucleus under proteotoxic stress conditions (Fig 1A and B). We originally showed that Hikeshi-mediated nuclear import of HSP70 is activated in response to heat shock (acute heat stress) in living cells (Fig 1A), but in this study, we noticed that nuclear HSP70 was reduced under nonstressed conditions in Hikeshi-KO cells (Fig 1C), suggesting that Hikeshi also mediates nuclear import of HSP70 under nonstressed conditions. Although the mechanism of HSP70 nuclear import activation in response to heat shock in living cells is currently unclear, we previously showed that Hikeshi mediates the nuclear import of HSP70 without raising a temperature (or proteotoxic stress) in an in vitro reconstituted nuclear transport assay (Kose et al, 2012). This indicates that Hikeshi possesses the ability to mediate the nuclear import of HSP70 under nonstressed conditions. It is plausible that there exist yet unknown factors in the cellular environment

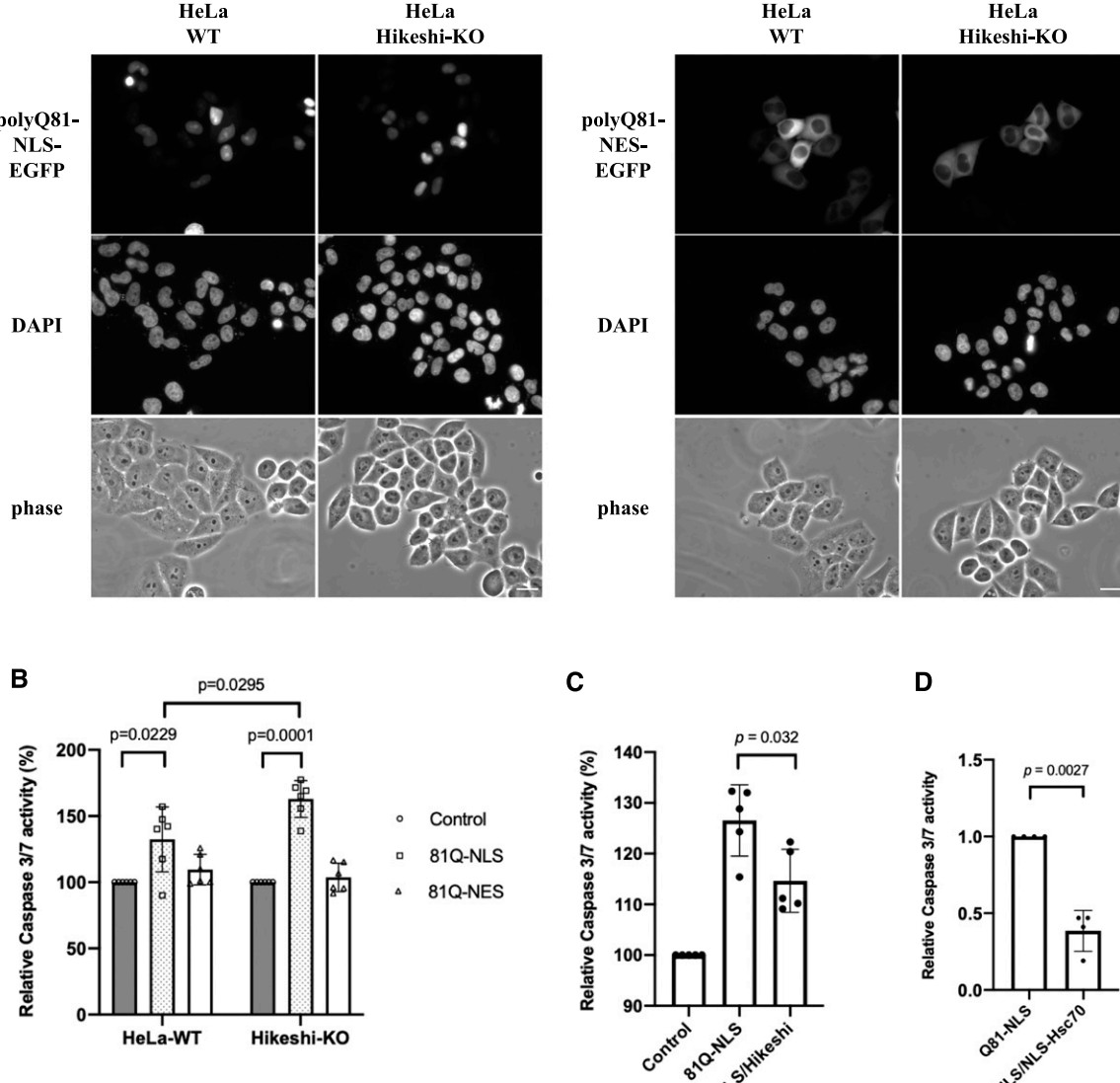

**A**

Human DRPLA (polyQ81)-NLS or NES-EGFP

5' -MVSTHHHHH- Q81 -HHGNSGPPRI- NLS or NES -GDPPVAT- EGFP - 3'

NLS:  PPKKKRKVEDP
NES:  L NSNELALKLAGLDINK

**Figure 5.  Hikeshi reduces nuclear polyQ-induced apoptosis.**
**(A)** Cellular localization of NLS or nuclear export signal (NES)-tagged polyQ81-EGFP. Plasmids expressing pathologic polyQ81 (from human DRPLA) fused to NLS or NES-EGFP proteins were transfected into WT and Hikeshi-KO HeLa cells. At 1 d posttransfection, cells were fixed, and fluorescence images of polyQ81-NLS-EGFP (left panels) and polyQ81-NES-EGFP (right panels) were captured with an Olympus BX51 microscope. DNA was counterstained with DAPI. **(B)** Hikeshi-KO cells expressing nuclear polyQ81 were more sensitive to apoptosis than WT cells. Cells were transfected with plasmids expressing EGFP (control), polyQ81-NLS-EGFP, or polyQ81-NES-EGFP. **(C, D)** Hikeshi and NLS-Hsc70 expression significantly reduces cellular apoptosis induced by nuclear polyQ81 protein expression. Hikeshi-KO cells were transfected with plasmids expressing EGFP (control) or polyQ81-NLS-EGFP with or without Hikeshi (C) or NLS-Hsc70 (D). In (B, C, D), at 2 d posttransfection, the induction of apoptosis by polyQ81 was measured by using the caspase-Glo 3/7 assay system (Promega). Error bars indicate ± SD. Statistical analyses were performed using Mann–Whitney two-tailed analysis.

responsible for regulating the efficiency of HSP70 nuclear import depending on the strength of stress, but such factors were lost in the in vitro assay.

RNA-seq analyses showed that 140 and 272 genes were up-regulated under nonstressed conditions in Hikeshi-KO HeLa or hTERT-RPE1 cells compared with WT cells (fold change > 1.5),

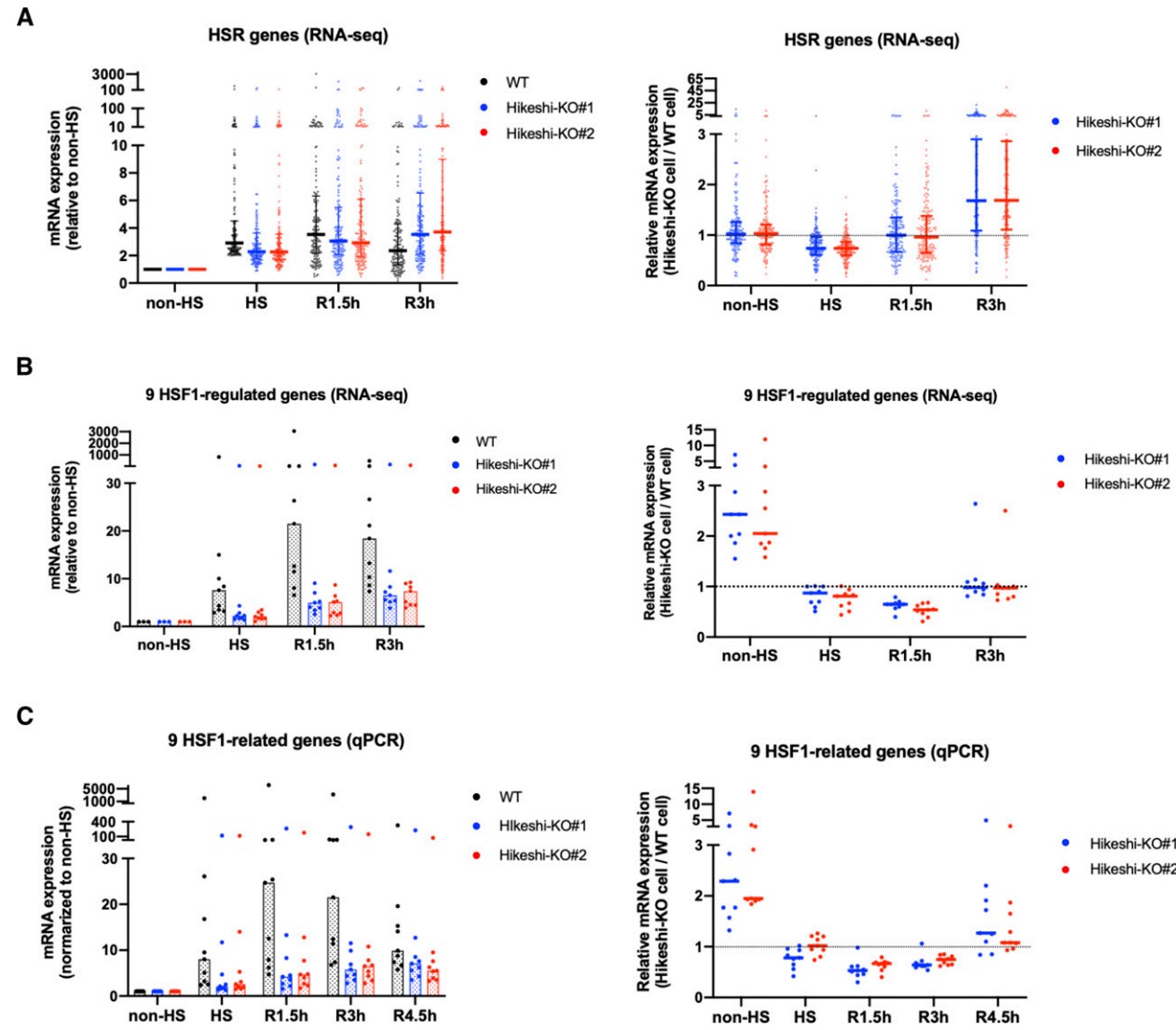

**Figure 6. Heat-shock response is impaired in the Hikeshi-KO cells.**
**(A)** mRNA expression levels of 162 heat-shock–responsive genes in HeLa WT and Hikeshi-KO cells. Genes whose expression level are induced more than twofold from normal to heat stress conditions in WT cells were categorized as heat-shock–responsive genes. Horizontal lines (in black, blue, red) show median with interquartile range. **(B, C)** mRNA expression levels of 9 HSF1-regulated genes, selected in Fig 2C, in HeLa WT and Hikeshi-KO cells. Bars (in black, blue, red; in left panels) and horizontal lines (in blue, red; in light panels) show median. In (A, B, C), mRNA expression levels relative to that under nonstressed (non-HS) conditions in each cell (left panels) or that in WT cells (right panels) were plotted. HeLa WT and Hikeshi-KO cells under non-HS conditions were exposed to heat shock (HS) at 43°C for 1 h and returned to nonstressed conditions at 37°C for 1.5 h (R1.5h), 3 h (R3h), and 4.5 h (R4.5h, in C). Gene expression at each time point was analyzed with RNA-seq (single replicate for each cell line, A, B) or qPCR (n = 3 biologically independent experiments, C).

respectively (Figs 2A and S3A and Table S1). GO analysis showed an enrichment of genes involved in protein folding, including the HSP70 chaperone family and its regulators (Figs 2B and S3B). Previous reports showed that many of these genes including nine genes we picked up in Figs 2C and S2C were regulated by HSF1 (Page et al, 2006; Vilaboa et al, 2017; Kovács et al, 2019). Furthermore, we confirmed that mRNA expressions of nine genes we picked up were strongly down-regulated in siHSF1-treated HeLa WT and Hikeshi-KO cells with qPCR (Fig 2D). These results suggest that HSF1 is chronically activated in Hikeshi-KO cells under nonstressed conditions. This activation is rather mild compared with the activation induced in response to heat shock (Fig S5A and B and S6A and B).

There are two main possible reasons for this: negative regulation of HSF1 activity is impaired, and chronic proteotoxic stress occurs in Hikeshi-KO cells.

Inhibition of nuclear import of HSP70 in Hikeshi-KO cells induced up-regulation of the transcriptional activity of HSF1 under non-stressed conditions (Figs 2C and 3D). The knowledge that HSP70 is the main regulator of HSF1 have become more evident from recent studies in yeast (Zheng et al, 2016; Krakowiak et al, 2018; Masser et al, 2019; Peffer et al, 2019). In mammalian cells, HSP70 and its cochaperone HSP40 was implicated in suppressing HSF1 activity in previous studies (Shi et al, 1998), and it was recently shown the detail mechanism that HSP70 binds to multiple sites in HSF and

remove HSF1 from DNA to restore the nonstressed state (Kmiecik et al, 2020). However, because HSP70 is largely localized in the cytoplasm under nonstressed conditions, it remains unclear whether nuclear HSP70 has any role in regulating HSF1 transcriptional activity under nonstressed conditions. In this study, we showed that depletion of Hikeshi resulted in the decrease of nuclear HSP70 (Fig 1C). Therefore, the interaction of HSP70 with HSF1 is likely to be severely restricted in the nuclei of Hikeshi-KO cells, resulting in the failure to negatively regulate HSF1. This situation leads to low chronic transcription of HSF1-regulated genes under nonstressed conditions in Hikeshi-KO cells. Conversely, an increase in nuclear-localized HSP70 effectively suppressed the mRNA expression of HSF1-regulated genes (Figs 3C and S4A and B) and the HSEs-driven reporter gene (Figs 3E and S4C) under the same conditions. Although the nucleocytoplasmic distribution of HSF1 is dynamic, a large fraction of mammalian HSF1 resides in the nucleus at a steady state under nonstressed conditions (Cotto et al, 1997; Jolly et al, 1997; Mercier et al, 1999; Vujanac et al, 2005). It makes sense that nuclear HSP70 regulates the activity of HSF1 in mammalian cells under nonstressed conditions. In HeLa cells, HSP70s (HSPA8/HSPA1A/HSPA1B) are calculated as $1.5 \times 10^7$ per cell, whereas HSF1 is $5.6 \times 10^4$ copies per cell (Kulak et al, 2014), resulting in a number of HSP70s being 270 times higher than that of HSF1 in HeLa cells. Thus, a small fraction of HSP70 localized in the nucleus is presumed to be sufficient for occupying HSF1 under nonstressed conditions. Analogous to the dissociation of HSF1 from DNA by HSP70 and HSP40, the conformation of the DNA-binding domain of p53 is regulated by the direct binding of HSP70 and HSP40 to p53 (Boysen et al, 2019; Dahiya et al, 2019). The nuclear HSP70 system may be committed to ensuring conformational and transcriptional regulation of several transcription factors.

Our observations showed that depletion of Hikeshi perturbs the appropriate heat-shock response in many ways. In response to heat shock, many of HSR genes and HSF1-regulated genes were weakly up-regulated in Hikeshi-KO cells than in WT cells at heat-shock (HS) period, but their mRNA expressions gradually continue to increased up to R3h (3 h after HS) in Hikeshi-KO cells, whereas mRNA expressions of HSR genes and HSF1-regulated genes in WT cells peaked at R1.5h and then decreased at R3h. Fig 6, suggesting that up-regulation of these gene in response to heat shock was delayed and sustained (and delayed shut-off) beyond the acute heat-shock period in Hikeshi-KO cells. In addition, the expression levels of many HSP genes including HSF1-regulated genes in Hikeshi-KO cells were higher than those in WT cells under nonstressed conditions, whereas the expression levels of the same genes in Hikeshi-KO cells were lower than those in WT cells at R1.5h and R3h (Fig S7). As described above, HSP70 itself is an important regulator of HSF1 activity, constituting a negative feedback loop that controls the heat-shock response (Abravaya et al, 1992; Baler et al, 1992; Shi et al, 1998; Zheng et al, 2016; Krakowiak et al, 2018; Masser et al, 2019; Peffer et al, 2019; Kmiecik et al, 2020). Because nuclear HSP70 is depleted in Hikeshi-KO cells, deactivation of HSF1 occurs inefficiently under nonstressed conditions and during recovery after heat shock in Hikeshi-KO cells. In addition, an absence of nuclear HSP70 is predicted to decrease the capacity of nuclear proteostasis. Taken together, appropriate nuclear accumulation of HSP70 mediated by Hikeshi during heat shock ensures a proper heat-shock

response and adaptation of nuclear proteostasis, after restoration of the normal cellular state.

Nuclear proteostasis is considered to be maintained by similar chaperone systems operating in the cytoplasm, although our understanding of proteostasis in the nucleus is still limited. The present results showed that nuclear, but not cytosolic, Fluc-monitoring proteins were more inactivated in Hikeshi-KO cells than in WT cells (Fig 4B and D), and the expression of NLS-Hsc70 proteins suppressed the inactivation of nuclear Fluc proteins in Hikeshi-KO cells (Fig 4C). These results suggest that deprivation of nuclear HSP70 reduces the nuclear capacity of protein quality control (PQC) in Hikeshi-KO cells. We note, however, that this lower capacity of nuclear PQC in Hikeshi-KO cells does not lead to a lethal effect on cells because depletion of Hikeshi did not affect cell growth under nonstressed conditions (Rahman et al, 2017). However, a lower capacity of PQC disrupts the normal heat-shock response (also discussed above). It is plausible that relatively mild proteotoxic stress may occur chronically in the nuclei of Hikeshi-KO cells, which sustains the activation of HSF1 activity under nonstressed conditions. Alternatively, nuclear proteostasis that descends in Hikeshi-KO cells may induce protein instability of proteins that include transcription factors, resulting in disorder of the expression of several regulated genes.

Many human neurodegenerative diseases are associated with the production and aggregation of proteins with expanded polyglutamine (polyQ) tracts in neurons, leading to neuronal dysfunction and cell death (reviewed in Lieberman et al [2019]). In mouse models, overexpression of HSP70 suppressed neuropathology (Jana et al, 2000; Cummings et al, 2001; Adachi et al, 2003), and in contrast, loss of HSP70 exacerbated pathogenesis (Wacker et al, 2009). In addition, HSP70 together with HSP40 was shown to prevent toxic aggregation of disease polyQ proteins (Muchowski et al, 2000; Schaffar et al, 2004; Lotz et al, 2010). As expected from these facts, our results showed that the expression of nuclear pathologic polyQ proteins had a higher level of cytotoxic effects in the Hikeshi-KO cells than in the WT cells (Fig 5B), which was reduced by expression of Hikeshi and NLS-tagged HSP70 in the Hikeshi-KO cells (Fig 5C and D). Intriguingly, in the same experiments, cytoplasmic expression of pathologic polyQ proteins did not show a significant effect on caspase 3/7 activity (Fig 5B), suggesting that the cytoplasm had a higher capacity to suppress the cellular toxicity of aggregate-prone proteins. The autophagy system is likely to contribute largely to the elimination of protein aggregates in the cytoplasm.

Our knowledge of PQC in the nucleus is relatively limited. In this study, we showed that knockdown of Hikeshi in human cells led to dysfunction of nuclear HSP70, resulting in impairment of the heat-shock response and nuclear proteostasis. Recent studies have provided new insights into proteostasis related to nuclear HSP70 under stress conditions. Misfolded proteins evolved during stress transiently associate with the liquid-like granular component phase of the nucleolus to prevent their aggregation (Frottin et al, 2019). HSP70 functions in refolding and releasing these misfolded proteins from the nucleolus during recovery from heat shock. In addition, HSP70 regulates the phase separation of TDP-43, which is related to neurodegenerative disease, and prevents conversion into solidification (Yu et al, 2021). We previously reported that dysfunction of Hikeshi in humans causes infant hypomyelinating

leukoencephalopathy (Edvardson et al, 2016; Vasilescu et al, 2017), which may be a result of a reduction in nuclear HSP70. Future studies need to elucidate the molecular mechanisms and functions of Hikeshi, presumably coupled with nuclear HSP70, in cells and development. In addition, cochaperone DNAJ proteins play a crucial role in the HSP70 chaperone system, including the regulation of polyQ aggregation (Hageman et al, 2010; Gillis et al, 2013; Ito et al, 2016; Rodríguez-González et al, 2020; Thiruvalluvan et al, 2020). It will be important to understand how DNAJs are closely associated with HSP70 in nuclear proteostasis.

# Materials and Methods

### Evolutionary profile

We generated the evolutionary profile based on the presence or absence of orthologs of Hikeshi. Orthologs of Hikeshi were obtained from Ortholog Group OG6_102809 in OrthoMCL DB version 6.5 (Fischer et al, 2011). We searched homologs of Hikeshi using jackhammer in HMMER 3.3.2 (Chen et al, 1985) against reference proteomes of UniProt (UniProt Consortium, 2021) if the organism was not included in OrthoMCL DB. We then performed orthology inference using reciprocal best hit methodology (Overbeek et al, 1999).

### Cells and cell cultures

Hikeshi-KO HeLa and hTERT-RPE1 cells were previously generated with the CRISPR–Cas9 system (Rahman et al, 2017). HeLa cells stably expressing EGFP-NLS-Hsc70 were previously established by transfection of the pEF5/FRT/EGFP-GW/NLS-Hsc70 plasmid (Rahman et al, 2017).

Cells were cultured at 37°C with 5% $CO_2$ in DMEM supplemented with 10% FBS. For heat-shock treatment, cells were incubated for 1 h at 43°C with prewarmed DMEM supplemented with 10% FBS and 20 mM HEPES (pH 7.3) in a water bath.

### Immunofluorescence

Cells were grown on coverslips coated with poly-L-lysine (Wako). After treatment with or without heat shock or MG132 as indicated in each figure caption, cells were washed with PBS (137 mM NaCl, 2.7 mM KCl, 8 mM $Na_2HPO_4$, and 1 mM $KH_2PO_4$), fixed in 2% formaldehyde/PBS (Polysciences) for 15 min at 37°C and then incubated with 50 mM glycine/HMK (20 mM HEPES [pH 7.5], 1 mM $MgCl_2$, and 100 mM KCl) for 5 min at room temperature. After permeabilization with 0.5% Triton X-100/HMK and blocking with 3% skim milk/PBS, cells were incubated with primary antibodies for 1 h at room temperature and detected with secondary antibodies conjugated with Alexa Fluor 488 or 594 (Thermo Fisher Scientific). Coverslips were mounted in PPDI (80% glycerol in PBS and 1 mg/ml paraphenylenediamine [11873580001; Roche]). DNA was counterstained with DAPI (Roche) or DRAQ5 (DR50050; Biostatus). Images were captured with an Olympus BX51 microscope (Figs 1A and B, 3A, and 5A) or FV1200 confocal microscope (Fig 1C).

The following antibodies were used as primary antibodies at the indicated dilutions: rat anti-Hsc70 (1:500; 1B5, Enzo Life Sciences) and mouse anti-Hsc70/Hsp70 (1:3,000; 1H5–1, Kose et al, 2012).

### RNA-seq and bioinformatics analysis

Total RNA was extracted from cultured cells with TRIzol reagent (Invitrogen) and further purified with an RNeasy Mini Kit and RNase-Free DNase Set (QIAGEN). After selection of poly-A, cDNA libraries were prepared with SureSelect Strand-Specific RNA Library Preparation Kits (Agilent) and subjected to 36-bp single-end sequencing with a HiSeq 2500 system (Illumina).

GO and gene enrichment analyses were performed with the Metascape online website (http://metascape.org/gp/index.html) (Zhou et al, 2019).

### Quantitative RT-PCR

RNA was extracted from cultured cells with TRIzol reagent (Invitrogen) and further purified with an RNeasy Mini Kit and RNase-Free DNase Set (QIAGEN). cDNA was synthesized using PrimeScript RT Master Mix (Takara), and qPCR was performed using Applied Biosystems 7500 Real-Time PCR System with TB Green Premix Ex Taq II (Tli RNaseH Plus) (Takara). Primers were selected from PrimerBank (Wang et al, 2012), and Primers for HSPA1A and HSPA1B were purchased from Takara. Quantification was performed using the $2^{-\Delta\Delta Ct}$ method, and expression was normalized to $\beta$-actin (ACTB). In experiments using siHikeshi-transfected cells and stably expressing EGFP-NLS-Hsc70 cells, expression was normalized to TATA binding protein (TBP) because expression levels of $\beta$-actin in RNA-seq analyses were relatively variable in these cells compared with control cells. Primer sequences are as follows:

ACTB (CATGTACGTTGCTATCCAGGC, CTCCTTAATGTCACGCACGAT)
BAG3 (ATTCCGGTGATACACGAGCAG, GCTGGTGGGTCTGGTACTC)
DEDD2 (TGAAGGCAAAGTGACCTGTGA, AGGCGTCCAGATAGGAGAGC)
DNAJA4 (GGGATGTTTATGACCAAGGCG, GCCAATTTCTTCGTGACTCCA)
DNAJB1 (CCAGTCACCCACGACCTTC, CCCTTCTTCACTTCGATGGTCA)
Hikeshi/C11orf73 (TAGGATTTGTCACGAATGGGAAG, AGCAACAGATGGAGTTCGGAC)
HSF1 (CCATGAAGCATGAGAATGAGGC, CTTGTTGACGACTTTCTGTTGC)
HSPA1A (AGCTGCTGCGACAGTCCACTAC, GTTCGCTCTGGGAAGCCTTG)
HSPA1B (GGTCAGGCCCTACCATTGAG, TCCTTGAGTCCCAACAGTCCA)
HSPA1L (CTACTGCCAAGGGAATCGCC, GCCGATCAGACGTTTAGCATCA)
HSPA6 (CATCGCCTATGGGCTGGAC, GGAGAGAACCGACACATCGAA)
HSPH1 (ACAGCCATGTTGTTGACTAAGC, GCATCTAACACAGATCGCCTCT)
TBP (CCACTCACAGACTCTCACAAC, CTGCGGTACAATCCCAGAACT)

### siRNA

Human Hikeshi (C11orf73) and HSF1 siRNAs were purchased from QIAGEN: siRNA-Hikeshi-2 (target sequence 5'-CAGCAAGUGGCA-GAGGAUAAA-3') (Kose et al, 2012) and siHSF1-3 (target sequence 5'-CAGGTTGTTCATAGTCAGAAT-3'). siRNA duplexes were transfected using the Lipofectamine RNAiMAX transfection reagent (Thermo Fisher Scientific).

## Immunoblotting

Cells were washed with ice-cold PBS and lysed with 2× Laemmli sample buffer. Cell lysates were subjected to SDS–PAGE and blotted onto PVDF membranes using the Trans-Blot Turbo Transfer System (Bio-Rad). After blocking with 10% skim milk/PBS containing 0.2% Tween-20, membranes were probed with primary mouse anti-Hsc70/Hsp70 (1:2,000; 1H5-1, Kose et al, 2012) and rabbit anti-Hikeshi antibodies (1:200; 20524-1-AP; Proteintech) and secondary goat anti-mouse or rabbit IgG conjugated to HRP (1:3,000; Bio-Rad) antibodies diluted in 10% skim milk/PBS containing 0.2% Tween-20. Images of chemiluminescent signals were detected and captured using ECL Western blotting detection reagents (Amersham) and FUSION Solo 7S (Vilber-Lourmat). Band intensities were measured with ImageJ software.

## Plasmid DNA constructions

An expression vector of the EGFP-NLS-Hsc70 protein, pEF5/FRT/EGFP-GW/NLS-Hsc70, was previously constructed (Kose et al, 2012). For construction of pEF5/FRT/EGFP-GW/NES-Hsc70 expressing EGFP-NES-Hsc70 proteins, the NLS region of the gateway entry vector pENTR/NLS-Hsc70 was converted to the NcoI-BamHI fragment of PKI NES (NSNELALKLAGLDINK) (Fung et al, 2015).

The gateway entry vector pENTR4β/Hikeshi was constructed by cloning human Hikeshi cDNA into the pENTR4 vector (Invitrogen). The gateway destination vector pcDNA-FLAG-GW was constructed by insertion of the FLAG-tagged sequence into the HindIII and ApaI sites of the pcDNA3.1/nV5-DEST vector (Invitrogen), converting the V5-tag to the FLAG-tag. The expression vector of FLAG-tagged Hikeshi pcDNA-FLAG-GW/Hikeshi was constructed with Gateway technology using pENTR4β/Hikeshi and pcDNS-FLAG-GW vectors.

For construction of the pcDNA-3xFLAG-EGFP vector, the HindIII-KpnI fragment of the 3xFLAG-tag was inserted into the pcDNA3.1/His vector (Invitrogen), converting the His-tag to the 3xFLAG-tag, and then the KpnI-BamHI fragment of EGFP was further inserted.

Gateway entry vectors, pENTR4β/Fluc-WT and SM, were constructed by cloning the modified coding region for the firefly luciferase gene from the pGL3-control vector (Promega) into the pENTR4 vector (Invitrogen). SM, an R188Q single mutation (Gupta et al, 2011), was introduced with basic PCR site-directed mutagenesis. Gateway entry vectors, pENTR4β/NLS-Fluc-WT and SM, were constructed by additional N-terminal insertion of the SV40 NLS sequence (PPKKKRKVEDP) into pENTR4β/Fluc-WT and SM vectors. Expression vectors of Fluc-WT/SM-EGFP and NLS-Fluc-WT/SM-EGFP with or without SV40 NLS, pcDNA-DEST53/Fluc-WT or SM and pcDNA-DEST53/NLS-Fluc-WT or SM were constructed with Gateway technology using each entry vector and pcDNA-DEST53 vector (Invitrogen).

For the expression vector of polyQ-NLS or NES-EGFP proteins, human DRPLA cDNA containing 81 CAG repeats (Onodera et al, 1997; Fujimoto et al, 2005) was subcloned into the XhoI and EcoRI sites of pEGFP-N1 (Clontech), and then the EcoRI-BamHI fragments of SV40 NLS (PPKKKRKVEDP) and PKI NES (NSNELALKLAGLDINK) (Fung et al, 2015) were inserted into the EcoRI and BamHI sites.

## Luciferase reporter gene assay

Cells were plated in 96-well plates at a density of $1 \times 10^4$ cells per 100 µl per well. After 24 h, cells were transfected with 110 ng of plasmid

DNA containing the pGL4.41[luc2P/HSE/Hygro] vector (Promega) per well using FuGENE HD (Promega). The pNL1.1TK (Nluc/TK) vector (Promega) was used as a transfection control. One day post-transfection, luciferase activities were measured using the luciferase assay reagent (Promega) or Nano-Glo Dual-Luciferase Reporter Assay (Promage), and then the luminescence was measured using GloMax 20/20 or GloMax navigator (Promega) (integration, 1 s).

## Luciferase assay

Fluc activities were measured using a luciferase assay system (Promega). Cells were plated in 96-well plates at a density of $1 \times 10^4$ cells per 100 µl per well. After 24 h, cells were transfected with 110 ng plasmid DNA per well using FuGENE HD (Promega). At 1 d posttransfection, cells were treated with 25 µg/ml cycloheximide for 2 h at 37°C (in Fig 4), washed with PBS, and then lysed with 20 µl/well of 1× cell culture lysis reagent (CCLR). A total of 10 µl of each cell lysate was mixed with 50 µl of luciferase assay reagent, and then the luminescence was measured using GloMax 20/20 or GloMax navigator (Promega) (integration, 1 s).

## Caspase-Glo 3/7 assay

Apoptosis was assessed by measuring caspase 3/7 activities using a caspase-Glo 3/7 assay (Promega). Cells were plated in white-walled 96-well plates (3917; Costar) at a density of $5 \times 10^3$ cells per 100 µl in each well. After 24 h, cells were transfected with 100 ng plasmid DNA per well using FuGENE HD (Promega). At 2 d posttransfection, 100 µl of caspase-Glo 3/7 reagent was added to each well. After incubation for 1 h at room temperature, the luminescence was measured using a GloMax navigator (Promega) according to the manufacturer's procedures.

# Data Availability

RNA-seq data have been deposited on DDBJ: DRA003217, DRA005851, and DRA013986.

# Supplementary Information

# Acknowledgements

We thank Y Ogawa for experiment with confocal microscopy, M Takagi for qPCR experiments, and members of Cellular Dynamics laboratory for their helpful discussions. DNA sequencing was performed by the Support Unit for Bio-Material Analysis, RIKEN BSI Research Resource Center. This work was supported by JSPS KAKENHI to S Kose (19K06500), K Imai (18K11543 and 21H03551), N Imamoto (15H05929, 18H02442, and 21H02482).

## Author Contributions

S Kose: conceptualization, formal analysis, funding acquisition, investigation, and writing—original draft, review, and editing.
K Imai: resources, formal analysis, and investigation.
A Watanabe: investigation.
A Nakai: resources and supervision.
Y Suzuki: resources, data curation, and investigation.
N Imamoto: conceptualization, supervision, funding acquisition, project administration, and writing—original draft, review, and editing.

## Conflict of Interest Statement

The authors declare that they have no conflict of interest.

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
