## [Reviewer comments · Life Science Alliance]

Life Science Alliance

Lack of Hikeshi activates HSF1 activity under normal conditions and disturbs the heat shock response

Shingo Kose, Naoko Imamoto, Kenichiro Imai, Ai Watanabe, Akira Nakai, and Yutaka Suzuki

DOI: <https://doi.org/10.26508/lsa.202101241>

Corresponding author(s): Naoko Imamoto, RIKEN; and Shingo Kose, RIKEN

Review Timeline:

Submission Date:	2021-09-22
Editorial Decision:	2021-10-25
Revision Received:	2022-04-01
Editorial Decision:	2022-04-21
Revision Received:	2022-04-27
Accepted:	2022-04-27

Transaction Report:

October 25, 2021

Re: Life Science Alliance manuscript #LSA-2021-01241-T

Dr. Naoko Imamoto
RIKEN
Cellular Dynamics Laboratory
2-1 Hirosawa
Wako, Saitama 351-0198
Japan

Dear Dr. Imamoto,

Thank you for submitting your manuscript entitled "Nuclear HSP70, supplied by Hikeshi, controls HSF1 activity and affects the heat stress response" to Life Science Alliance. The manuscript was assessed by expert reviewers, whose comments are appended to this letter. We invite you to submit a revised manuscript addressing the Reviewer comments.

Thank you for this interesting contribution to Life Science Alliance. We are looking forward to receiving your revised manuscript.

Sincerely,

B. MANUSCRIPT ORGANIZATION AND FORMATTING:

Reviewer #1 (Comments to the Authors (Required)):

Kose et. al., explore nuclear roles of the ubiquitous chaperone protein Hsp70, a prominent protein in stress responses, under non-stressed conditions. The researchers link Hikeshi, a nuclear Hsp70 membrane transporter protein, to the proper regulation of genes under control of the stress response transcription factor Hsf1, which is negatively regulated by Hsp70. This paper uses Hikeshi knockouts to show nuclear localization dependence of Hsp70 on Hikeshi, and proteostasis aberrations when that transport is interrupted. Experiments with artificially targeted nuclear Hsp70 (NLS-Hsp70) are convincing and bolster the findings. Overall, the quality of the work is high and complements recent findings in the yeast model system. Several issues should be addressed to improve the study.

- 1) Fig. 1B would benefit from a DAPI stain or other positive control for nuclear localization. The claim of nuclear localization in 1C in response to MG132 is less convincing and the results should be modified to reflect minor localization relative to heat shock.
- 2) Fig. 2D - there is some variability in HSR derepression resulting from CRISPR knockout vs. siRNA depletion of Hikeshi - did the authors quantify Hikeshi levels in the latter experiments by western blot? It is important to do so when using shRNA or siRNA approaches.
- 3) Fig. 3E - did the authors consider titrating "regular" Hsp70 without the NLS attached? This could help reveal if Hsp70 is capable of functionally repressing HSF1 in an Hikeshi null background. Given the results of Fig. 1D showing significant Hsp70 localization in Hikeshi mutants, additional pathway(s) must be involved.
- 4) Fig. 4D - The legend references Fluc protein stability in wild type or knockout cells, but the Y axis of the plot is labeled "activity". Which was measured? GFP signal or luciferase activity? Only the former reflects protein stability. The latter is a more complicated property that reflects possible changes in stability and folding. Given the minor differences shown in Fig. 4B, these considerations are important.
- 5) What is happening in Fig. 6C - why are heat shock genes downregulated in the knockout lines after heat shock? This result seems contrary to the previous results. Additionally, Fig. 6G should be moved to supplementary information and/or summarized in a more easily interpretable manner.
- 6) No citations describing the major advancements in understanding Hsp70 regulation of HSF1 in yeast via nuclear Hsp70 are included. Given that very little has been done in this area in vertebrate cells and much has been revealed in yeast, this is an unacceptable oversight. The following references should be included as part of the discussion regarding the Hsp70/Hsf1 regulatory circuit:

1. Masser, A. E., Kang, W., Roy, J., Kaimal, J. M., Quintana-Cordero, J., Friedländer, M. R., and Andréasson, C. (2019) Cytoplasmic protein misfolding titrates Hsp70 to activate nuclear Hsf1. *Elife*. 10.7554/eLife.47791
2. Peffer, S., Gonçalves, D., and Morano, K. A. (2019) Regulation of the Hsf1-dependent transcriptome via conserved bipartite contacts with Hsp70 promotes survival in yeast. *J. Biol. Chem.* 294, 12191-12202
3. Krakowiak, J., Zheng, X., Patel, N., Feder, Z. A., Anandhakumar, J., Valerius, K., Gross, D. S., Khalil, A. S., and Pincus, D. (2018) Hsf1 and Hsp70 constitute a two-component feedback loop that regulates the yeast heat shock response. *Elife*. 10.7554/eLife.31668
4. Zheng, X., Krakowiak, J., Patel, N., Beyzavi, A., Ezike, J., Khalil, A. S., and Pincus, D. (2016) Dynamic control of Hsf1 during heat shock by a chaperone switch and phosphorylation. *Elife*. 10.7554/eLife.18638

Reviewer #2 (Comments to the Authors (Required)):

This study from an eminent laboratory is disappointing in many key aspects, the most important being that it does not directly address in any way the question of mechanism i.e. the study largely uses Hikeshi knockout cells and tries to draw conclusions

about HSF1, but does not attempt to modulate HSF1 (eg. by siRNA) to try to confirm observations. Many of the results are not only overinterpreted, but also lack confirmation of the "screening" results for differential expression by qPCR, and there are no details regarding how many replicates are used, how many separate experiments were performed etc. Overall, this study is too preliminary to be considered for publication in its present form.

Specific Points

1. Hsp70 is strongly excluded from the nucleus in untreated WT cells In Fig 1d but localises to some extent in the nucleus in untreated WT cells in Fig 1b. Why is Hsp70 localisation in Fig 1b and Fig 1d so different ? This is the first figure of the manuscript - what is the correct result ? Can the authors be confident that the phenomena they are trying to document are reproducible or significant ?
2. Figure 2 is lacking in meaningful detail - eg. it reports 140 genes upregulated by Hikeshi KO, but how many were downregulated ? All differentially expressed genes resulting from Hikeshi KO under normal non stressed conditions should be listed. Were any of these results validated by PCR ? To what extent can the authors be confident the results are significant without validation ?
3. A trend in the entire paper is the lack of detail regarding replicates eg. the expression data in Fig 2c and 2d (and Fig EV1 c and d) are representative of the RNA seq results - how many replicates were performed for RNA seq in Figure 2 and Fig EV1? As per point #2, expression data in Fig 2c and 2d and Fig EV1 c and d needs to be validated by qPCR to give confidence that the results are consistent and significant. In Figure 3 it is not clear what the replicate size is for RNA sequencing (why is there no confirmation of the differential expression presented in Fig 3c by qPCR). In short, every figure needs clear information regarding the number of replicates, number of experiments - the impression, without this, is that everything was performed only once ? And further, confirmation of "screening" results by qPCR seems essential to give weight to the observational data.
4. The authors overstate/overinterpret their results on many occasions, and fail to perform qPCR to confirm conclusions of differential expression. Eg.
 - a. The title is misleading - it should read "Lack of Hikeshi reduces HSF1 activity and impacts the heat shock response"
 - b. On page 7 "depletion of Hikeshi affects the expression of various genes and particularly induces upregulation of mRNA expression transcribed from HSF1" should read "...particularly induces upregulation of expression of known HSF1 targets" because this specific data they are discussing (Fig 2) only shows that specific genes are upregulated by Hikeshi KO (and appear to just happen to be known targets of HSF1 - where are the references ?). It would be important for the authors to perform HSF1 knockdown in Hikeshi-KO to confirm the upregulation of these specific genes is HSF1 dependent.
 - c. The same issue arises in the discussion on page 13 of Fig 2b where the authors state "predictably, many of these Hsp70-related genes were upregulated by HSF1". This is not a correct description of the result, which did not validate the role of HSF1 in the upregulation of these genes. The authors really need to perform HSF1 knockdown in Hikeshi-KO to confirm the upregulation of these specific genes is indeed HSF1 dependent.
 - d. The abstract is misleading when it says "depletion of Hikeshi induces a reduction in nuclear HSP70 and upregulation of the mRNA expression of genes regulated by HSF1 under nonstressed conditions". The authors have not confirmed HSF1 upregulated these genes in their experimental system, with the results essentially indirect (ie. reduced activation of the heat shock element promoter in Hikeshi KO cells). At best the authors could argue that there may be a correlation between lack of Hikeshi, and lower HSF1 activity, but if the authors performed HSF1 knockdown in Hikeshi-KO to confirm the upregulation of these specific genes, they would be able to make claims that the effects are HSF1 dependent.
 - e. On page 8, "Therefore, dysfunction of Hikeshi, which leads to a decrease in nuclear Hsp70, causes dysregulation of HSF1 transcriptional activity" should read "Therefore, lack of Hikeshi, which leads to a decrease in nuclear Hsp70..."
 - f. Figure 5 legend title states "Hikeshi suppresses nuclear polyQ-induced apoptosis"; in fact, Hikeshi downregulates OR reduces OR contributes to suppression of nuclear polyQ-induced apoptosis because we can see that caspase activity is still increased in WT cells, albeit at a lower level than in KO cells (Fig 5b). Similarly page 10 "..KO cells significantly suppressed the apoptosis induction..."
5. On page 11, the authors conclude that "the heat shock response was sustained during recovery from heat shock" in the Hikeshi-KO cells because more genes that are heat shock responsive were at their peak expression levels 3 h after stress in the Hikeshi KO cell compared to WT cells. It seems much more likely that the transcriptional response to heat shock was delayed, with peak expression of heat shock responsive genes occurring 3 hours after heat stress ?
6. The authors suggest Hikeshi imports Hsp70 under non stressed conditions but have not shown binding interaction under non stressed conditions.
7. The authors should acknowledge that given that many heat shock responsive genes show delayed upregulation rather than inhibited upregulation in the Hikeshi-KO cells (Fig 6d). Can the authors be sure there is not another factor compensating for loss of Hikeshi to affect transcription of these genes ?

Minor Points

1. Figure 1a should go into the supplementary figures. The whole paragraph "Hikeshi orthologs are widely distributed in eukaryote" should be removed from results and put into introduction or discussion.
2. A number of spelling mistakes should be amended including those in legend Figure 3 and Figure 4
3. On page 8 the authors state Fluc is frequently used to monitor chaperone activity of Hsp70. The authors need to supply some references here.
4. The authors' claim on P. 7 that "many of these genes that are upregulated in Hikeshi-KO under nonstressed conditions are known to be regulated by ... HSF-1" needs appropriate literature citations (or the text should be deleted).
5. The authors go backwards and forwards between heat shock response and heat stress response - are these the same ? What is meant by the different terms ? Can the authors be clear ?

In short, the current paper requires a complete overhaul in terms not only of the writing, but also requires a major body of confirmatory experimentation, including HSF1 knockdown experiments. Only in this way, can the study be considered solid or rigorous enough for Life Science Alliance.

Reviewer #3 (Comments to the Authors (Required)):

Comments on «Nuclear HSP70, supplied by Hikeshi, controls HSF1 activity and affects the heat stress response»

In the article entitled «Nuclear HSP70, supplied by Hikeshi, controls HSF1 activity and affects the heat stress response», Kose et al. investigate the function of Hikeshi in non-stressed as well as heat stressed cells in regulating gene expression and proteostasis. With this work, they follow up on their previous papers, in which they showed that Hikeshi functions as a nuclear import factor for HSP70 upon heat stress. Here they now show that Hikeshi is also required to maintain a steady state level of nuclear HSP70 in unstressed cells. Lack of Hikeshi leads to changes in gene expression, notable higher expression of several HSF1 target genes. This effect could be rescued by HSP70 targeted to the nucleus, suggesting that the misregulation is due to the failure of Hikeshi to import HSP70. The authors further show a positive effect of nuclear localized HSP70 on preventing aggregate formation of a polyQ protein, stabilization of an unstable Luciferase reporter and suppressing gene misregulation observed in Hikeshi ko cells.

Overall, this paper presents a number of interesting observations but some additional control experiments and some changes in the presentation of the data are needed before publication.

Major paper conclusions:

1. "Knockout of the Hikeshi gene induces a reduction in nuclear HSP70 under nonstressed conditions"
 - Conclusion well supported by Figure 1.
2. "Knockout of the Hikeshi gene induces upregulated expression of the HSF1-regulated gene under nonstressed conditions"
 - Conclusion partially supported by data presented in Figure 2. A detailed description of how the analysis of the RNAseq data was performed is missing. Were there replicates? What are the statistical test used to select the upregulated genes? I am no expert in mRNAseq but it seems that the information given here is insufficient and I cannot judge whether the analysis done is appropriate. As an example, the text states for figure 2D that "Consistently, siRNA-mediated depletion of Hikeshi also showed that most of these HSF1-regulated genes, except for HSPA1L, were upregulated under nonstressed conditions", however for BAG3 this increase does not look significant.
3. "Nuclear HSP70 regulates the transcriptional activity of HSF1"
 - Conclusion largely supported by data presented in Figure 3. As control for Figure 3B,E and 4C, co-transfection of a Hsc70 plasmid with NES should be shown to exclude indirect effects of overexpression not related to nuclear localization.
4. "Nuclear HSP70 functions in the protein stability of nuclear luciferase monitoring proteins."
 - Conclusion largely supported by data presented in Figure 4. As control for Figure 4C, co-transfection of a Hsc70 plasmid with NES should be shown to exclude indirect effects of overexpression not related to nuclear localization.
5. "Hikeshi suppresses nuclear polyQ-induced apoptosis."
 - Conclusion largely supported by Figure 5: Is the effect of polyQ also alleviated if NLS-HSP70 is expressed? This would again exclude indirect effects.
6. "Heat shock response is impaired in the Hikeshi-KO cells."
 - This is not entirely convincing. Figure 6C: if in unstressed conditions the levels are different, than fold change heat stress versus normal growth will be different even if the absolute expression level is identical. Is absolute induction reduced in HikeshiKO? Figure 6G would suggest that the levels are actually similar in heat stress.
 - Figure 6D,E: very difficult to read this chart! The concept of "delayed genes" is rather difficult to understand for me and it is also unclear, what the delayed peak actually means. I guess it suggests that there is continued transcription that leads to accumulation of the mRNA beyond the acute heat shock period, but this does not really mean a delay in expression but rather a delay in shut-off. I'm not sure how to improve this section, but maybe the authors can come up with a more intuitive way of presenting their message.
 - 6G: In the discussion it says that in HS most cells in KO cells were lower expressed, but this is obvious only in the recovery, not in the HS conditions. This does not clearly fit with the argumentation that the peak is later, or does it? Levels are similar in HS but reduced in recovery would actually suggest faster shut off or decay?

Minor comments

1. "supplied by Hikeshi" is an unusual formulation that I found difficult to understand in the title. I would suggest to change this to e.g. "imported by" or "Hikeshi-imported nuclear HPS70"

2. The introduction would profit from a few additional citations:

- o Exogenously expressed classical nuclear localization signal (NLS)-tagged HSP70, which is carried into the nucleus by the importin- α/β pathway, significantly suppresses cell death of Hikeshi-depleted HeLa cells caused by heat stress.

- o Its transcriptional activity of HSF1 is regulated by various posttranslational modifications, including phosphorylation.

3. p5: "that Hikeshi was acquired only after the emergence" I'm not an expert here, but I doubt that acquisition of Hikeshi after the emergence of eukaryotic cells alone is enough evidence to implicate a function connected to the nucleus.

4. Section title p6: "upregulated expression of the HSF1-regulated gene" should be "upregulated expression of HSF1-regulated genes"

5. Were the cell lines tested for mycoplasma? In the DAPI stain in Figure 5 there are some small dots visible around the cells.

Point-by-point response

Reviewer Comments:

Reviewer #1 (Comments to the Authors (Required)):

Kose et. al., explore nuclear roles of the ubiquitous chaperone protein Hsp70, a prominent protein in stress responses, under non-stressed conditions. The researchers link Hikeshi, a nuclear Hsp70 membrane transporter protein, to the proper regulation of genes under control of the stress response transcription factor Hsf1, which is negatively regulated by Hsp70. This paper uses Hikeshi knockouts to show nuclear localization dependence of Hsp70 on Hikeshi, and proteostasis aberrations when that transport is interrupted. Experiments with artificially targeted nuclear Hsp70 (NLS-Hsp70) are convincing and bolster the findings. Overall, the quality of the work is high and complements recent findings in the yeast model system. Several issues should be addressed to improve the study.

1) Fig. 1B would benefit from a DAPI stain or other positive control for nuclear localization. The claim of nuclear localization in 1C in response to MG132 is less convincing and the results should be modified to reflect minor localization relative to heat shock.

We added images of DAPI staining in new Figs 1A and 1B.

To describe differences between the effect of MG132 and heat stress for nuclear accumulation of HSP70, we have rewritten the text (Results: “Knockout of the Hikeshi gene induces a reduction in nuclear HSP70 under nonstressed conditions”, p5, paragraph 2) as follows.

“In accordance with previous reports (Kose et al, 2012; Rahman et al, 2017), heat shock induced nuclear import of HSP70 was strongly inhibited in Hikeshi-KO cells (Fig 1A). The addition of MG132, a proteasome inhibitor well known to induce cellular proteotoxic stress, also induced nuclear accumulation of HSP70, but to lesser extent than those of heat shock. The addition of MG132 also induced significant nucleolar accumulation of HSP70, which was also seen in cells under heat shock conditions. These MG132 induced nuclear/nucleolar accumulation of HSP70 was strongly inhibited in the Hikeshi-KO cells (Fig 1B). The effect of Hikeshi KO on heat shock and MG132 induced nuclear/nucleolar accumulation of HSP70 was similarly seen in HeLa cells and hTERT-RPE1 cells. “

2) Fig. 2D - there is some variability in HSR derepression resulting from CRISPR knockout vs. siRNA depletion of Hikeshi - did the authors quantify Hikeshi levels in the latter experiments by western blot? It is important to do so when using shRNA or siRNA approaches.

We quantified protein levels of Hikeshi by western blotting, and the results is shown in new Fig S2C. The protein expression levels of Hikeshi were reduced to ~10% by siRNA treatment. Quantitative real-time PCR analysis showed that mRNA expression levels of Hikeshi were reduced to about 5 % by siRNA treatment. In addition to this, we described a variability in HSR derepression resulting from CRISPR knockout vs. siRNA depletion of Hikeshi in Text (Results: “Knockout of the Hikeshi gene induces a reduction in nuclear HSP70 under nonstressed conditions”, p7, paragraph 1) as follows.

“siRNA-mediated depletion of Hikeshi, by which protein expression levels of Hikeshi were reduced to ~10%, also showed that most of these 9 genes were upregulated under nonstressed conditions, with some exceptions (Fig S2C). We found some variability in upregulation levels of these 9 genes resulting from Hikeshi-KO cells and siHikeshi-treated cells. Unlike RNA-seq results with Hikeshi-KO cells (Fig 2C), in siHikeshi-treated cells, HSPA1L was not upregulated in RNA-seq experiment, BAG3 was not significantly upregulated in both RNA-seq and qPCR experiments, and DEDD2 was not upregulated in qPCR experiment (Fig S2C). We presume these discrepancies due to protein levels of Hikeshi remaining in siHikeshi-treated cells.”

3) Fig. 3E - did the authors consider titrating "regular" Hsp70 without the NLS attached? This could help reveal if Hsp70 is capable of functionally repressing HSF1 in an Hikeshi null background. Given the results of Fig. 1D showing significant Hsp70 localization in Hikeshi mutants, additional pathway(s) must be involved.

We performed the reporter assay using non-tagged Hsc70. The results are shown in new Fig. S4C. Contrary to the inhibitory effect of NLS-Hsc70, expression of non-tagged Hsc70 or NES-tagged Hsc70 enhanced luciferase activities in the Hikeshi-KO cells in a dose-dependent manner, which show that HSP70 that localized in the nucleus inhibited transcriptional activation through the HSE promoter. These results are mentioned in Text (Results: “Nuclear HSP70 regulates the transcriptional activity of HSF1”, p9, paragraph 2) as follows.

“The increased luciferase activities in the Hikeshi-KO cells were strongly suppressed by cotransfection with the NLS-Hsc70-expressing plasmid in a dose-dependent manner (Fig 3E). Cotransfection of non-tagged or nuclear export signal (NES)-tagged Hsc70 expressing plasmid

did not suppress the luciferase activity (Fig S4C). These results show that HSP70 that localized in the nucleus inhibited transcriptional activation through the HSE promoter.”

The enhanced effect on luciferase activities of non-tagged or NES-tagged Hsc70 may indicate that cytoplasmic luciferases that expressed through the HSE promoter are stabilized by chaperone function of cytoplasmic Hsc70.

4) Fig. 4D - The legend references Fluc protein stability in wild type or knockout cells, but the Y axis of the plot is labeled "activity". Which was measured? GFP signal or luciferase activity? Only the former reflects protein stability. The latter is a more complicated property that reflects possible changes in stability and folding. Given the minor differences shown in Fig. 4B, these considerations are important.

We measured luciferase activities after inhibition of the newly-synthesized luciferase proteins by treatment with cycloheximide, as described in Text (Results: Hikeshi-mediated nuclear-localized HSP70 functions in nuclear protein stability, p 9, paragraph 3). The Y axis of the plot is “luciferase activity” instead of GFP signal. As Reviewer’s comment, we agree that luciferase activity is influenced by both protein stability and re-folding activity of chaperones. Nevertheless, we consider that we can claim that Fluc protein “stability” is affected by nuclear Hsp70, because luciferase activity of nuclear-localized Fluc-SM, a single mutant (SM) protein that are more unstable than wild-type Fluc, was more influenced than nuclear-localized Fluc-WT in the Hikeshi-KO cells (Fig 4B). It is likely that absence of nuclear Hsp70s influence protein stability. We added words to describe this in Text (Results: Hikeshi-mediated nuclear-localized HSP70 functions in nuclear protein stability, p 9, paragraph 3 to p10, paragraph 1) as below.

“Notably, these effects were more significant with the NLS-Fluc-R188Q single mutant (SM), which tends to be more unstable than wild-type Fluc (Gupta et al, 2011). Although luciferase activity is influenced by both protein stability and re-folding activity of chaperones, our data showing that luciferase activity of nuclear-localized Fluc-SM was more influenced than that of nuclear-localized Fluc-WT in the Hikeshi-KO cells may suggest that the function of nuclear Hsp70 confers with protein stability.”

To avoid confusion, we changed “protein stability” to “protein activity” of legend title in Figure 4.

5) What is happening in Fig. 6C - why are heat shock genes downregulated in the knockout lines after heat shock? This result seems contrary to the previous results. Additionally, Fig. 6G should be moved to supplementary information and/or summarized in a more easily interpretable manner.

I am sorry for this confusion. HSF1-regulated genes are weakly upregulated in Hikeshi KO cells compared with WT cells under normal conditions (previous and present Fig 2C), while their heat-shock-responsive expressions (fold change, from nonstressed to heat shock conditions) are suppressed in Hikeshi KO cells compared to WT cells (previous Fig 6C). In accordance with the reviewer's suggestions, and to avoid confusions, we reorganized Figure 6, including new data of qPCR, and completely rewrote this section in Text (Results: Heat shock response is impaired in the Hikeshi-KO cells; p 11, paragraph 2 to p12, paragraph 3).

In new Fig 6, we show that the expression levels of heat-shock-responsive (HSR) genes (defined as genes whose expressions are induced for more than 2-fold at heat shock conditions relative to nonstressed conditions in WT cells) are upregulated in response to heat shock even in Hikeshi-KO cells but to lesser extent than WT cells (new Fig 6A). 9 HSF1 regulated genes we picked up in Fig 2C (shown in previous Fig 6C) showed similar expression tendency with HSR genes as shown in new Fig 6B and C. The results in new Fig 6 showed that expression levels of HSR- and HSF1-regulated genes were suppressed in the Hikeshi-KO cells compared with that in WT cells under heat shock condition, which suggest that heat shock response is impaired in the Hikeshi KO cells.

We moved previous Fig 6G, which showed mRNA expression levels of heat shock proteins (HSPs) family genes in Hikeshi-KO cells relative to that in WT cells, to new supplemental Fig S7 and further added new graph to visualize the result of Table in Fig S7.

6) No citations describing the major advancements in understanding Hsp70 regulation of HSF1 in yeast via nuclear Hsp70 are included. Given that very little has been done in this area in vertebrate cells and much has been revealed in yeast, this is an unacceptable oversight. The following references should be included as part of the discussion regarding the Hsp70/Hsf1 regulatory circuit:

*1. Masser, A. E., Kang, W., Roy, J., Kaimal, J. M., Quintana-Cordero, J., Friedländer, M. R., and Andréasson, C. (2019) Cytoplasmic protein misfolding titrates Hsp70 to activate nuclear Hsf1. *Elife*. 10.7554/eLife.47791*

*2. Peffer, S., Gonçalves, D., and Morano, K. A. (2019) Regulation of the Hsf1-dependent transcriptome via conserved bipartite contacts with Hsp70 promotes survival in yeast. *J. Biol. Chem.* 294, 12191-12202*

3. Krakowiak, J., Zheng, X., Patel, N., Feder, Z. A., Anandhakumar, J., Valerius, K., Gross, D. S., Khalil, A. S., and Pincus, D. (2018) *Hsf1 and Hsp70 constitute a two-component feedback loop that regulates the yeast heat shock response. Elife. 10.7554/eLife.31668*

4. Zheng, X., Krakowiak, J., Patel, N., Beyzavi, A., Ezike, J., Khalil, A. S., and Pincus, D. (2016) *Dynamic control of Hsf1 during heat shock by a chaperone switch and phosphorylation. Elife. 10.7554/eLife.18638*

Thank you for these suggestions. We cited all these manuscripts in “Results” and “Discussion” regarding the HSP70/HSF1 regulatory circuit (p 8, paragraph 1; p14, paragraph 2; p15, paragraph 1). We rewrote the sentences in Text (Discussion: p 14, paragraph 2) as below.

“The knowledge that HSP70 is the main regulator of HSF1 have become more evident from recent studies in yeast (Zheng et al, 2016; Krakowiak et al, 2018, Peffer et al, 2019, Masser et al, 2019). In mammalian cells, HSP70 and its cochaperone HSP40 was implicated in suppressing HSF1 activity in previous studies (Shi et al, 1998), and it was recently shown the detail mechanism that HSP70 binds to multiple sites in HSF and remove HSF1 from DNA to restore the nonstressed state (Kmiecik et al, 2020).”

Reviewer #2 (Comments to the Authors (Required)):

This study from an eminent laboratory is disappointing in many key aspects, the most important being that it does not directly address in any way the question of mechanism i.e. the study largely uses Hikeshi knockout cells and tries to draw conclusions about HSF1, but does not attempt to modulate HSF1 (eg. by siRNA) to try to confirm observations. Many of the results are not only overinterpreted, but also lack confirmation of the "screening" results for differential expression by qPCR, and there are no details regarding how many replicates are used, how many separate experiments were performed etc. Overall, this study is too preliminary to be considered for publication in its present form.

Specific Points

1. Hsp70 is strongly excluded from the nucleus in untreated WT cells In Fig 1d but localises to some extent in the nucleus in untreated WT cells in Fig 1b. Why is Hsp70 localisation in Fig 1b and Fig 1d so different ? This is the first figure of the manuscript - what is the correct result ? Can the authors be confident that the phenomena they are trying to document are reproducible or significant ?

We used different microscopes. In new Fig 1 A and B, images were captured with an Olympus BX51 microscope. In new Fig C, images were captured with a FV1200 confocal microscope. The apparent differences in images are caused by use of different microscope.

We quantified the nuclear and cytoplasmic intensities of Hsc70 under nonstressed conditions in WT and Hikeshi-KO cells in new Fig 1A. As shown Fig A below (Figure presented only for reviewers), levels of nuclear HSP70 detected by immunofluorescence under normal conditions was lower in Hikeshi-KO cells compared with WT cells, and are compatible with images taken by confocal microscope (see new Fig 1C). We are confident in the reproducibility of our data.

To avoid confusion, the microscope we used was indicated in figure legend of Fig 1 and Material and Methods (Immunofluorescence, p 18, paragraph 1).

Fig A

2. Figure 2 is lacking in meaningful detail - eg. it reports 140 genes upregulated by Hikeshi KO, but how many were downregulated? All differentially expressed genes resulting from Hikeshi KO under normal non stressed conditions should be listed. Were any of these results validated by PCR? To what extent can the authors be confident the results are significant without validation?

mRNA expression of 155 genes was downregulated in Hikeshi-KO HeLa cells compared with WT HeLa cells under nonstressed conditions (fold change < 0.7). We showed these results and Gene ontology term analysis in new Figure S2 A&B and added the list of upregulated and downregulated genes in a new supplementary Table S1 excel file. These were described in Text (Results: Knockout of the Hikeshi gene induces upregulated expression of HSF1-regulated genes under nonstressed conditions; p 6, paragraph 2).

To validate results of RNA-seq data, we performed quantitative real-time PCR analyses and added these results in new Figures (2C, 2D, S2C, S3C, S3D, S4A, S4B, and S5).

3. A trend in the entire paper is the lack of detail regarding replicates eg. the expression data in

Fig 2c and 2d (and Fig EV1 c and d) are representative of the RNA seq results - how many replicates were performed for RNA seq in Figure 2 and Fig EV1? As per point #2, expression data in Fig 2c and 2d and Fig EV1 c and d needs to be validated by qPCR to give confidence that the results are consistent and significant. In Figure 3 it is not clear what the replicate size is for RNA sequencing (why is there no confirmation of the differential expression presented in Fig 3c by qPCR). In short, every figure needs clear information regarding the number of replicates, number of experiments - the impression, without this, is that everything was performed only once ? And further, confirmation of "screening" results by qPCR seems essential to give weight to the observational data.

We agree with these comments. We performed one RNA-seq experiment for each cell and picked up genes whose expression level change were common to two different knockout cells lines (Hikeshi KO #1 and Hikeshi KO #2 for HeLa cell, Hikeshi KO #3 and Hikeshi KO #4 for hTERT-RPE1 cell) as shown in Fig 2, Fig S2, Fig 3, and Fig S3. In our revised manuscript, we performed qPCR analyses (n=3 biologically independent samples) to validate results of RNA-seq and added these results in new Figs (2C, 2D, S2C, S3C, S3D, S4A, S4B, and S5). Statistical significance was determined using unpaired t-test (NS; no significance, $p > 0.05$).

qPCR results confirmed that the expression levels of all 9 genes, which we selected as HSF1-regulated genes (in RNA seq data), were significantly upregulated (in new Fig 2C). In accordance with the reviewer's comments, we further performed HSF1 knockdown experiments in WT and Hikeshi-KO cells to confirm whether these selected 9 genes are regulated by HSF1. qPCR results showed that mRNA expression levels of these genes were downregulated by HSF1 depletion, confirming that all these genes are indeed regulated by HSF1. We added these results in new Fig 2D and described in Text (Result: Knockout of the Hikeshi gene induces upregulated expression of HSF1-regulated genes under nonstressed conditions; p 7, paragraph 1).

On the other hand, in qPCR experiments, we obtained different results from RNA-seq data in some genes, which are described in Text (p 7 – p 8). For example, qPCR analysis showed that HSPA1L, but not DEDD2, was significantly upregulated in siHikeshi-treated HeLa cells, whereas RNA-seq analysis showed that DEDD2, but not HSPA1L, was upregulated (new Fig S2C). Meanwhile, RNA-seq analysis showed that mRNA expression of HSPA1L was suppressed in HeLa cells stably-expressing NLS-Hsc70, whereas qPCR analysis showed that mRNA expression of HSPA1L was not significantly suppressed in one of the two clonal cells stably-expressing NLS-Hsc70 and HeLa cells transiently-expressing NLS-Hsc70 (new Figs S4A, B). mRNA expression levels of HSPA1L, which is known to be highly expressed in the testis, under normal conditions in HeLa and RPE1 cells is very low (RNA-seq data in new Fig

S5). This may make it difficult to assess HSPA1L expression with qPCR technique.

In summary, we conclude that results of qPCR basically support our RNA-seq data, although there are only a few differences, We have rewritten our manuscript to include these qPCR results.

4. The authors overstate/overinterpret their results on many occasions, and fail to perform qPCR to confirm conclusions of differential expression. Eg.

a. The title is misleading - it should read "Lack of Hikeshi reduces HSF1 activity and impacts the heat shock response"

We changed the title as follows:

“Lack of Hikeshi activates HSF1 activity under normal conditions and disturbs the heat shock response”

b. On page 7 "depletion of Hikeshi affects the expression of various genes and particularly induces upregulation of mRNA expression transcribed from HSF1" should read "...particularly induces upregulation of expression of known HSF1 targets" because this specific data they are discussing (Fig 2) only shows that specific genes are upregulated by Hikeshi KO (and appear to just happen to be known targets of HSF1 - where are the references ?). It would be important for the authors to perform HSF1 knockdown in Hikeshi-KO to confirm the upregulation of these specific genes is HSF1 dependent.

c. The same issue arises in the discussion on page 13 of Fig 2b where the authors state "predictably, many of these Hsp70-related genes were upregulated by HSF1". This is not a correct description of the result, which did not validate the role of HSF1 in the upregulation of these genes. The authors really need to perform HSF1 knockdown in Hikeshi-KO to confirm the upregulation of these specific genes is indeed HSF1 dependent.

d. The abstract is misleading when it says "depletion of Hikeshi induces a reduction in nuclear HSP70 and upregulation of the mRNA expression of genes regulated by HSF1 under nonstressed conditions". The authors have not confirmed HSF1 upregulated these genes in their experimental system, with the results essentially indirect (ie. reduced activation of the heat shock element promoter in Hikeshi KO cells). At best the authors could argue that there may be a correlation between lack of Hikeshi, and lower HSF1 activity, but if the authors performed HSF1 knockdown in Hikeshi-KO to confirm the upregulation of these specific genes, they would be able to make claims that the effects are HSF1 dependent.

We performed HSF1 knockdown in WT and Hikeshi KO cells, and performed qPCR experiments of 9 genes that we claimed as HSF1-regulated genes. The qPCR results showed that mRNA expression levels of all 9 genes were downregulated by HSF1 depletion, indicating that mRNA expression of 9 genes are regulated by HSF1. The results are shown in new Fig 2D, and described in the Text (Results: Knockout of the Hikeshi gene induces upregulated expression of the HSF1-regulated gene under nonstressed conditions; p 7, paragraph 1). We also cited the following papers for describing HSF1 target genes (p 6, paragraph 2; p 7, paragraph 1; p 13, paragraph 3).

Page TJ, Sikder D, Yang L, Pluta L, Wolfinger RD, Kodadek T, Thomas RS (2006) Genome-wide analysis of human HSF1 signaling reveals a transcriptional program linked to cellular adaptation and survival. *Mol Biosyst* 2: 627-639

Vilaboa N, Boré A, Martin-Saavedra F, Bayford M, Winfield N, Firth-Clark S, Kirton SB, Voellmy R (2017) New inhibitor targeting human transcription factor HSF1: effects on the heat shock response and tumor cell survival. *Nucleic Acids Res* 45: 5797-5817

Kovács D, Sigmond T, Hotzi B, Bohár B, Fazekas D, Deák V, Vellai T, Barna J (2019) HSF1Base: A Comprehensive Database of HSF1 (Heat Shock Factor 1) Target Genes. *Int J Mol Sci* 20: 5815

e. On page 8, "Therefore, dysfunction of Hikeshi, which leads to a decrease in nuclear Hsp70, causes dysregulation of HSF1 transcriptional activity" should read "Therefore, lack of Hikeshi, which leads to a decrease in nuclear Hsp70..."

We rewrote “dysfunction of Hikeshi” to “lack of Hikeshi”. (p 9, paragraph 2).

f. Figure 5 legend title states "Hikeshi suppresses nuclear polyQ-induced apoptosis"; in fact, Hikeshi downregulates OR reduces OR contributes to suppression of nuclear polyQ-induced apoptosis because we can see that caspase activity is still increased in WT cells, albeit at a lower level than in KO cells (Fig 5b). Similarly page 10 "...KO cells significantly suppressed the apoptosis induction..."

We rewrote “suppress” to “reduce” in legend of Figure 5, and in p11 paragraph1 “Coexpression of Hikeshi (Fig 5C) or NLS-Hsc70 (Fig 5D) in the Hikeshi-KO cells significantly reduced the apoptosis induction caused by nuclear polyQ81 proteins”

5. On page 11, the authors conclude that "the heat shock response was sustained during

recovery from heat shock" in the Hikeshi-KO cells because more genes that are heat shock responsive were at their peak expression levels 3 h after stress in the Hikeshi KO cell compared to WT cells. It seems much more likely that the transcriptional response to heat shock was delayed, with peak expression of heat shock responsive genes occurring 3 hours after heat stress ?

We have reanalyzed the experimental data in Fig 6. As pointed out by this reviewer, the result showed that the transcriptional response to heat shock was delayed in the Hikeshi-KO cells, and then mRNA expressions of these genes continued to increase during 3 hours after heat shock. We therefore agree with reviewer's comment. We described results shown in new Fig 6 in Text (Results: Heat shock response is impaired in the Hikeshi-KO cell; p 11 paragraph 3 to p12 paragraph 2) including our comments "These results suggest that in the Hikeshi-KO cells, upregulation of HSR genes in response to heat shock is weakened and is sustained at HS and beyond (up to 3 hrs after recovery to normal temperature after HS)"

6. The authors suggest Hikeshi imports Hsp70 under non stressed conditions but have not shown binding interaction under non stressed conditions.

We previously showed that Hikeshi interacts efficiently with ATP-bound HSP70 in the pulldown assay without raising temperature and mediates nuclear import of HSP70 in an in vitro transport assay without raising the temperature (Kose et al, 2012). We have stated these previous observations in the Discussion (page 13, paragraph 2).

7. The authors should acknowledge that given that many heat shock responsive genes show delayed upregulation rather than inhibited upregulation in the Hikeshi-KO cells (Fig 6d). Can the authors be sure there is not another factor compensating for loss of Hikeshi to affect transcription of these genes ?

As described above (Q5), we have removed previous Fig 6D and completely reorganized the Fig 6. As pointed out by this reviewer, the result showed upregulation of heat-shock-responsive (HSR) genes in response to heat shock was weakened and delayed in Hikeshi-KO cells. We will acknowledge a possibility that we cannot exclude another factor compensating effect of Hikeshi on transcription of many of these genes.

Minor Points

1. Figure 1a should go into the supplementary figures. The whole paragraph "Hikeshi orthologs are widely distributed in eukaryote" should be removed from results and put into introduction or discussion.

According to reviewer's suggestion, we moved previous Fig 1A to new supplementary Fig S1, and previous first session of Results was described in the Introduction of revised manuscript (page 4, paragraph 3).

2. A number of spelling mistakes should be amended including those in legend Figure 3 and Figure 4

Thank you for this. We corrected spelling mistakes including those in legend of Figure 3 and Figure 4.

3. On page 8 the authors state Fluc is frequently used to monitor chaperone activity of Hsp70. The authors need to supply some references here.

We added the following citations for use of Fluc to monitor chaperone activity of Hsp70 in Text (p 9, paragraph 3).

Schröder H, Langer T, Hartl FU, Bukau B (1993) DnaK, DnaJ and GrpE form a cellular chaperone machinery capable of repairing heat-induced protein damage. *EMBO J* 12: 4137-4144

Frydman J, Nimmesgern E, Ohtsuka K, Hartl FU (1994) Folding of nascent polypeptide chains in a high molecular mass assembly with molecular chaperones. *Nature* 370: 111-117

Terada K, Kanazawa M, Bukau B, Mori M (1997) The human DnaJ homologue dj2 facilitates mitochondrial protein import and luciferase refolding. *J Cell Biol* 139: 1089-1095

4. The authors' claim on P. 7 that "many of these genes that are upregulated in Hikeshi-KO under nonstressed conditions are known to be regulated by HSF-1" needs appropriate literature citations (or the text should be deleted).

We added the following citations for showing that genes, which were picked up in Fig 2C, were regulated by HSF1 in Text (p 6, paragraph 2; p 7, paragraph 1; p 13, paragraph 3).

Page TJ, Sikder D, Yang L, Pluta L, Wolfinger RD, Kodadek T, Thomas RS (2006)

Genome-wide analysis of human HSF1 signaling reveals a transcriptional program linked to cellular adaptation and survival. *Mol Biosyst* 2: 627-639

Vilaboa N, Boré A, Martin-Saavedra F, Bayford M, Winfield N, Firth-Clark S, Kirton SB, Voellmy R (2017) New inhibitor targeting human transcription factor HSF1: effects on the heat shock response and tumor cell survival. *Nucleic Acids Res* 45: 5797-5817

Kovács D, Sigmond T, Hotzi B, Bohár B, Fazekas D, Deák V, Vellai T, Barna J (2019) HSF1Base: A Comprehensive Database of HSF1 (Heat Shock Factor 1) Target Genes. *Int J Mol Sci* 20: 5815.

5. The authors go backwards and forwards between heat shock response and heat stress response - are these the same ? What is meant by the different terms ? Can the authors be clear ?

We meant heat shock as “acute heat stress”, therefore, we unified to heat stress to heat shock in the revised manuscript.

In short, the current paper requires a complete overhaul in terms not only of the writing, but also requires a major body of confirmatory experimentation, including HSF1 knockdown experiments. Only in this way, can the study be considered solid or rigorous enough for Life Science Alliance.

We have confirmed major body of confirmatory experiment as described above, and revised writing as suggested by this reviewer.

Reviewer #3 (Comments to the Authors (Required)):

Comments on «Nuclear HSP70, supplied by Hikeshi, controls HSF1 activity and affects the heat stress response»

In the article entitled «Nuclear HSP70, supplied by Hikeshi, controls HSF1 activity and affects the heat stress response», Kose et al. investigate the function of Hikeshi in non-stressed as well as heat stressed cells in regulating gene expression and proteostasis. With this work, they follow up on their previous papers, in which they showed that Hikeshi functions as a nuclear import factor for HSP70 upon heat stress. Here they now show that Hikeshi is also required to maintain a steady state level of nuclear HSP70 in unstressed cells. Lack of Hikeshi leads to changes in gene expression, notable higher expression of several HSF1 target genes. This effect could be

rescued by HSP70 targeted to the nucleus, suggesting that the misregulation is due to the failure of Hikeshi to import HSP70. The authors further show a positive effect of nuclear localized HSP70 on preventing aggregate formation of a polyQ protein, stabilization of an unstable Luciferase reporter and suppressing gene misregulation observed in Hikeshi ko cells. Overall, this paper presents a number of interesting observations but some additional control experiments and some changes in the presentation of the data are needed before publication.

Major paper conclusions:

1. "Knockout of the Hikeshi gene induces a reduction in nuclear HSP70 under nonstressed conditions"

• Conclusion well supported by Figure 1.

2. "Knockout of the Hikeshi gene induces upregulated expression of the HSF1-regulated gene under nonstressed conditions"

• Conclusion partially supported by data presented in Figure 2. A detailed description of how the analysis of the RNAseq data was performed is missing. Were there replicates? What are the statistical test used to select the upregulated genes? I am no expert in mRNAseq but it seems that the information given here is insufficient and I cannot judge whether the analysis done is appropriate. As a example, the text states for figure 2D that "Consistently, siRNA-mediated depletion of Hikeshi also showed that most of these HSF1-regulated genes, except for HSPA1L, were upregulated under nonstressed conditions", however for BAG3 this increase does not look significant.

We performed one RNA-seq experiment for each cell and picked up genes whose expression levels change were similar in two different knockout cell lines. In the revised manuscript, to validate results of RNA-seq, we performed qPCR analyses (n=3 biologically independent samples) and added these results in new Figs (2C&D, 6C, S2C, S3C&D, S4A&B, S6B). Statistical significance was determined using unpaired t-test (NS; no significance, $p > 0.05$).

As pointed out by reviewer, qPCR analysis showed that BAG3 (and DEDD2) was not significantly upregulated in siHikeshi-treated HeLa cells, whereas significant upregulation of HSPA1L was confirmed by qPCR, but not previous RNA-seq. We presume some variability in upregulation levels of these genes resulting from Hikeshi-KO cells and siHikeshi-treated cells due to protein levels of Hikeshi remaining in siHikeshi-treated cells (We quantified protein levels of Hikeshi by western blotting, new Fig S2C). We described about it in Text (p 7, paragraphs 1) and removed previous Fig 2D to supplemental Fig S2C.

Taken together, we think that qPCR results basically support our RNA-seq data, although there are only a few differences. We have updated our manuscript to include qPCR

results in Text (Three sessions of Results: page 6-7, page 8, and page 12).

3. "Nuclear HSP70 regulates the transcriptional activity of HSF1"

• Conclusion largely supported by data presented in Figure 3. As control for Figure 3B,E and 4C, co-transfection of a Hsc70 plasmid with NES should be shown to exclude indirect effects of overexpression not related to nuclear localization.

4. "Nuclear HSP70 functions in the protein stability of nuclear luciferase monitoring proteins."

• Conclusion largely supported by data presented in Figure 4. As control for Figure 4C, co-transfection of a Hsc70 plasmid with NES should be shown to exclude indirect effects of overexpression not related to nuclear localization.

As suggested by this reviewer, we performed the reporter assay using NES-tagged Hsc70. Contrary to the inhibitory effect of NLS-Hsc70, expression of NES-tagged (and non-tagged) Hsc70 did not inhibit but rather enhanced luciferase activities in the Hikeshi-KO cells in a dose-dependent manner. We think, this result indicates that cytoplasmic luciferases expressed through the HSE promoter are stabilized by chaperone function of cytoplasmically-expressed Hsc70. We showed these results in new Fig S4C and described in the Text (p 9, paragraph 1).

Regarding Fig 4C, we performed the assay using NES-tagged Hsc70. Hikeshi-KO cells were co-transfected with 60 ng of plasmids expressing NLS-Fluc-WT or SM and 50 ng plasmids expressing NLS- or NES-Hsc70. Fluc activities were measured after treatment with or without cycloheximide at 1day post transfection. The result is shown in the Fig B below. Unexpectedly, NES-tagged Hsc70 also increased luciferase activity of NLS-Fluc. However, NLS-Hsc70 promoted NLS-Fluc activity more efficiently in all three experiments (NLS-Fluc-WT and SM activities were higher in NLS-Hsc70-transfected cells than that in NES-Hsc70-transfected cells). In particular, NLS-Hsc70 efficiently suppressed the decrease of more unstable mutant NLS-Fluc-SM. Overexpression of cytoplasmic Hsc70 (NES-Hsc70) might increase stability of cytoplasmic NLS-Fluc prior to its nuclear localization, like NES-Hsc70 showed positive effect on cytoplasmic Fluc activity in new Fig S4C.

Fig B

5. *"Hikeshi suppresses nuclear polyQ-induced apoptosis."*

• *Conclusion largely supported by Figure 5: Is the effect of polyQ also alleviated if NLS-HSP70 is expressed? This would again exclude indirect effects.*

According to suggestions, we performed the experiment using NLS-tagged Hsc70. Expression of NLS-Hsc70, as well as Hikeshi, suppressed nuclear polyQ-induced apoptosis. We added these results in new Fig 5D, and described in Text (p11, paragraph 1).

6. *"Heat shock response is impaired in the Hikeshi-KO cells."*

• *This is not entirely convincing. Figure 6C: if in unstressed conditions the levels are different, than fold change heat stress versus normal growth will be different even if the absolute expression level is identical. Is absolute induction reduced in HikeshiKO? Figure 6G would suggest that the levels are actually similar in heat stress.*

• *Figure 6D,E: very difficult to read this chart! The concept of "delayed genes" is rather difficult to understand for me and it is also unclear, what the delayed peak actually means. I guess it suggests that there is continued transcription that leads to accumulation of the mRNA beyond the acute heat shock period, but this does not really mean a delay in expression but rather a delay in shut-off. I'm not sure how to improve this section, but maybe the authors can come up with a more intuitive way of presenting their message.*

We agree with these comments. Therefore, we completely reorganized Figure 6 including new data of qPCR. We removed previous Fig 6A, B, D-F, and moved Fig 6G to new supplemental Fig S7.

In new Fig 6A, we showed RNA-seq data on mRNA expression levels of 162 heat-shock-responsive (HSR) genes, which we defined as genes whose expression level are induced more than 2-fold from normal conditions in response to heat shock in WT cells. In left panel, to show heat stress response in each cell, mRNA expression levels at HS and beyond (R1.5h and R3h) relative to that under non-stressed (non-HS) conditions in each cell were plotted. In right panel, to compare absolute expression levels between WT and Hikeshi-KO cells at each time point, mRNA expression levels in Hikeshi-KO cells relative to that in WT cells were plotted.

As in case of the HSR genes, mRNA expression levels of 9 HSF1-regulated genes, selected in Fig 2C, were plotted in new Figs 6B (RNA-seq results) and 6C (qPCR results). mRNA expression levels of these genes at HS and beyond relative to that of non-HS in each cell (in left panel), or that at each time point in Hikeshi-KO cells relative to that in WT cells (in right

panel) were shown.

As shown in Fig 6A left panels, median of mRNA expression levels of HS relative to non-HS in two Hikeshi-KO cells are 2.28 and 2.27, while median of mRNA expression levels of HS relative to non-HS of WT cells is 2.91. Further, absolute mRNA expression levels of HSR genes at HS were lower in the Hikeshi-KO cells compared with that in WT cells (Fig 6A right panel). HSF1-regulated genes showed similar tendency (Fig 6B and C). Therefore, we concluded that heat-shock-responsive upregulation of these HSR and HSF1-regulated genes was weakened in Hikeshi-KO cells.

As for mRNA expressions beyond HS, mRNA expressions of HSR and HSF1-regulated genes in WT cells peaked at R1.5h, and then decreased at R3h. On the other hand, mRNA expressions of their genes in two Hikeshi-KO cells gradually increased from non-HS to R3h (left panels in Fig 6A-C). Consequentially, absolute mRNA expression of these genes in Hikeshi-KO cells tend to increase gradually beyond HS at higher levels than that in WT cells (right panels in Fig 6A-C).

Furthermore, our data in new Figs S5 (RNA-seq data) and S6 (qPCR data) showed that the maximum expressions of HSF1-regulated genes were not higher in two Hikeshi-KO cells than that in WT cells, at least during the observation period. (In Fig S5B RNA-seq data, median of maximum expression was 489 rpkm at R1.5h and R3h in WT cells, and 410 rpkm and 356 rpkm at R3h in KO#1 and KO#2, respectively. In Fig S6B qPCR data, median of maximum relative RNA expression was 24.7-fold at R1.5h in WT cells, 18.8-fold and 14.8-fold at R3h in KO#1 and KO#2, respectively).

Taken together, consistent with the reviewer's comment, we concluded that upregulation of these gene in response to heat shock was delayed and sustained (and delayed shut-off) beyond the acute heat shock period in Hikeshi-KO cells.

• 6G: In the discussion it says that in HS most cells in KO cells were lower expressed, but this is obvious only in the recovery, not in the HS conditions. This does not clearly fit with the argumentation that the peak is later, or does it? Levels are similar in HS but reduced in recovery would actually suggest faster shut off or decay?

We thank the reviewer for pointing out this. We moved previous Fig 6G to new supplemental Fig S7.

mRNA expression levels of heat shock proteins (HSPs) listed in the previous Fig 6G (new Fig S7) under heat stress conditions was similar among WT and Hikeshi-KO cells (right panel in new Fig S7). We rewrote the Text (Discussion: p15, paragraph 1) as follows.

“In addition, the expression levels of many heat shock protein (HSP) genes including HSF1-regulated genes in Hikeshi-KO cells were higher than those in WT cells under nonstressed conditions, while the expression levels of the same genes in Hikeshi-KO cells were lower than those in WT cells at R1.5h and R3h (Fig S7).”

Further, mRNA expression pattern of the HSF1-regulated genes we analyzed and the other HSP genes, which are not highly upregulated at HS (fold change < 2), seems to be different. We need further experiments to understand the mechanism of transcriptional regulation of these HSP genes in detail.

Minor comments

1. *"supplied by Hikeshi" is an unusual formulation that I found difficult to understand in the title. I would suggest to change this to e.g. "imported by" or "Hikeshi-imported nuclear HPS70"*

We changed the title as follow:

“Lack of Hikeshi activates HSF1 activity under normal conditions and disturbs the heat shock response”

2. *The introduction would profit from a few additional citations:*

o Exogenously expressed classical nuclear localization signal (NLS)-tagged HSP70, which is carried into the nucleus by the importin- α/β pathway, significantly suppresses cell death of Hikeshi-depleted HeLa cells caused by heat stress.

We added the following citation (Introduction: p 3, paragraph 1)

Kose S, Furuta M, Imamoto N (2012) Hikeshi, a nuclear import carrier for Hsp70s, protects cells from heat shock-induced nuclear damage. *Cell* 149: 578-589

o Its transcriptional activity of HSF1 is regulated by various posttranslational modifications, including phosphorylation.

We added the following citations (Introduction: p 3, paragraph 2).

Guettouche T, Boellmann F, Lane WS, Voellmy R. (2005) Analysis of phosphorylation of human heat shock factor 1 in cells experiencing a stress. *BMC Biochem* 6:4

Zheng X, Krakowiak J, Patel N, Beyzavi A, Ezike J, Khalil AS, Pincus D. (2016) Dynamic control of Hsf1 during heat shock by a chaperone switch and phosphorylation. *Elife* 5: e18638

Gomez-Pastor R, Burchfiel ET, Thiele DJ. (2018) Regulation of heat shock transcription factors and their roles in physiology and disease. *Nat Rev Mol Cell Biol* 19: 4-19

3. p5: "that Hikeshi was acquired only after the emergence" I'm not an expert here, but I doubt that acquisition of Hikeshi after the emergence of eukaryotic cells alone is enough evidence to implicate a function connected to the nucleus.

We moved Fig 1A to new supplementary Fig S1, and removed previous first session of "Results" and added it in Introduction (p 4, paragraph 3), according to the reviewer's comment.

4. Section title p6: "upregulated expression of the HSF1-regulated gene" should be "upregulated expression of HSF1-regulated genes"

We changed section title to "Knockout of the Hikeshi gene induces upregulated expression of HSF1-regulated genes under nonstressed conditions" (page 6).

5. Were the cell lines tested for mycoplasma? In the DAPI stain in Figure 5 there are some small dots visible around the cells.

We checked contamination of mycoplasma using MycoStrip (Mycoplasma Detection Kit, InvivoGen), and then confirmed that these cells were not contaminated with mycoplasma. We often saw such a DAPI staining when we transfected plasmid DNA into culture cells using FuGENE HD transfection reagent (Promega). We think that the appearance of such dots is due to experimental method using non-liposomal transfection reagents. Transfection reagent (FuGENE HD) we used in transfection is described in Materials and Methods.

April 21, 2022

RE: Life Science Alliance Manuscript #LSA-2021-01241-TR

Prof. Naoko Imamoto
RIKEN
Cellular Dynamics Laboratory
2-1 Hirosawa
Wako, Saitama 351-0198
Japan

Dear Dr. Imamoto,

Thank you for submitting your revised manuscript entitled "Lack of Hikeshi activates HSF1 activity under normal conditions and disturbs the heat shock response". We would be happy to publish your paper in Life Science Alliance pending final revisions necessary to meet our formatting guidelines.

- please address the final remaining Reviewer 2 and 3 points
- please add ORCID ID for secondary corresponding author-they've received instructions on how to do so
- please add the Twitter handle of your host institute/organization as well as your own or/and one of the authors in our system
- please make sure the author order in your manuscript and our system match
- please use the [10 author names, et al.] format in your references (i.e. limit the author names to the first 10)
- please add a callout for figure S1 & Figure S5B to your main manuscript
- please add the RNA seq deposition details (with access code) in the Data Availability section

A. FINAL FILES:

B. MANUSCRIPT ORGANIZATION AND FORMATTING:

Sincerely,

Reviewer #1 (Comments to the Authors (Required)):

The authors have adequately addressed my major concerns with additional data and convincing rebuttals.

Reviewer #2 (Comments to the Authors (Required)):

Overall, the paper is stronger after the inclusion of confirmatory qPCR/KD data. However, it requires additional revision/attention to detail in description of the data and methods.

Author Response to Our Revision 1

1. The authors claim that use of two different microscopes is responsible for the difference in the measured nucleus/cytoplasm ratio for the original Figure 1 (although the cells and antibody used are the same), but the question remains as to which result is more accurate. Perhaps the simplest is that the authors decide which microscopic approach is more precise, and present only data for that approach ?, unless the other microscopic analysis is providing a different insight ? - ideally, the primary quantification data for the nucleus/cytoplasm ratio (nucleus/cytoplasm/background) should be included (supp. Data perhaps ?).
3. RNA seq in Figure 2 was performed once (i.e. single replicates) for each cell line (WT, KO #1 and KO#2?), but is this the case for all of the other RNA seq data ? Please include this information for each of the figure legends that include RNA seq data.

Other Points

1. On page 5 the authors write that "The effect of Hikeshi KO on heat shock and MG132 induced nuclear/nucleolar accumulation of Hsp70 was seen in HeLa and htert-rpe1 cells" but the data figure is not referenced - can the authors show the data in a figure, or else provide a published reference?
2. There are no p values or "NS" for Figure 2 ?
3. Is there a significant increase in Fluc activity (Fig S4c) resulting from transfection of Hikeshi KO cells with the non-tagged Hsc70 or the NES tagged Hsc70 ? Can the authors discuss why should this be the case ?
4. In the results section "Heat shock response is impaired in Hikeshi-KO cells" on page 11, authors state 162 genes are upregulated more than 2-fold in response to heat shock stress. Please clarify in the text, are these the genes that are upregulated in the sample treated with stress and without any recovery?
5. In the results section "Heat shock response is impaired in Hikeshi-KO cells", the authors use the phrase "slightly upregulated" multiple times. Do the authors mean "small but not statistically significant upregulation" ?

Minor Points

1. Some figure legends in the figure panels are very hard to read ie. the dots that indicate what each sample is are too small and difficult to distinguish. This includes those in Figure 6a and many others. Please make sure all figures are sufficiently reader friendly and easy to distinguish such as those in Figure 6b.
2. In the last paragraph of the results section Figure S6 is cited but should be Figure S7 ?

Reviewer #3 (Comments to the Authors (Required)):

The authors have addressed my previous concerns and I judge this manuscript suitable for publication.

minor text edit:

Figure legend 2D: unclear formulation "normalized to that in each cell transfected with siGL2"

Point-by-point response

Reviewer Comments:

Reviewer #1 (Comments to the Authors (Required)):

The authors have adequately addressed my major concerns with additional data and convincing rebuttals.

Reviewer #2 (Comments to the Authors (Required)):

Overall, the paper is stronger after the inclusion of confirmatory qPCR/KD data. However, it requires additional revision/attention to detail in description of the data and methods.

Author Response to Our Revision 1

1. The authors claim that use of two different microscopes is responsible for the difference in the measured nucleus/cytoplasm ratio for the original Figure 1 (although the cells and antibody used are the same), but the question remains as to which result is more accurate. Perhaps the simplest is that the authors decide which microscopic approach is more precise, and present only data for that approach ?, unless the other microscopic analysis is providing a different insight ? - ideally, the primary quantification data for the nucleus/cytoplasm ratio (nucleus/cytoplasm/background) should be included (supp. Data perhaps ?).

As suggested by the reviewer, we added quantification data for the nucleus/cytoplasmic ratio of Hsc70 under nonstressed conditions in WT and Hikeshi-KO cells to Fig 1A. We consider both microscopic analysis is equally accurate.

3. RNA seq in Figure 2 was performed once (i.e. single replicates) for each cell line (WT, KO #1 and KO#2?), but is this the case for all of the other RNA seq data ? Please include this information for each of the figure legends that include RNA seq data.

We obtained all RNA-seq data from single replicate experiments. According to reviewer's comments, we added the sentence "single replicate for each cell line" in each of the figure legends (Fig 2, Fig 3, Fig 6, Fig S2, Fig S3, Fig S5, and Fig S7).

Other Points

1. On page 5 the authors write that "The effect of Hikeshi KO on heat shock and MG132 induced nuclear/nucleolar accumulation of Hsp70 was seen in HeLa and htert-rpe1 cells" but the data figure is not referenced - can the authors show the data in a figure, or else provide a published reference?

We added the following citation and added the sentence (see Rahman et al 2017, for under heat shock conditions) (page 5, paragraph 2).

Rahman KMZ, Mamada H, Takagi M, Kose S, Imamoto N (2017) Hikeshi modulates the proteotoxic stress response in human cells: Implication for the importance of the nuclear function of HSP70s. *Genes Cells* 22: 968-976

2. There are no p values or "NS" for Figure 2 ?

qPCR confirmed that all of 9 genes were significantly upregulated (in Fig 2C) or downregulated (in Fig 2D). We added *p* values ($***p < 0.001$, $**p < 0.01$) in Fig 2C (right panels) and D.

3. Is there a significant increase in Fluc activity (Fig S4c) resulting from transfection of Hikeshi KO cells with the non-tagged Hsc70 or the NES tagged Hsc70 ? Can the authors discuss why should this be the case ?

As discussed in response to Reviewer #1 in the 1st revision, we speculate that non-tagged or NES-tagged Hsc70 functions in protein stability and refolding of cytoplasmic luciferases expressed through the HSE promoter, resulting in the enhanced effect on luciferase activities. We rewrote the text (page 9, paragraph 1) as follows.

“Cotransfection of non-tagged or nuclear export signal (NES)-tagged Hsc70-expressing plasmid did not suppress, but rather increased the luciferase activity (Fig S4C). The enhanced effect on luciferase activities of non-tagged or NES-tagged Hsc70 may indicate that cytoplasmic luciferases that expressed through the HSE promoter are stabilized by chaperone function of cytoplasmic Hsc70.”

4. In the results section "Heat shock response is impaired in Hikeshi-KO cells" on page 11, authors state 162 genes are upregulated more than 2-fold in response to heat shock stress. Please clarify in the text, are these the genes that are upregulated in the sample treated with stress and without any recovery?

To avoid confusion, we added the sentence “treated with heat shock and without any recovery” (page 11, paragraph 2) as follows:

“In WT HeLa cells treated with heat shock and without any recovery, mRNA expression of 162 genes was induced more than 2-fold under heat shock conditions relative to nonstressed conditions. We categorized these 162 genes as heat-shock-responsive (HSR) genes.”

5. In the results section "Heat shock response is impaired in Hikeshi-KO cells", the authors use the phrase "slightly upregulated" multiple times. Do the authors mean "small but not statistically significant upregulation" ?

We apologize for our obscure description. We did not perform statistical analyses for RNA-seq data. To avoid confusion, we rewrote the sentences in Text (page 12, paragraph 3) as below.

“Furthermore, RNA-seq data showed that more than half of the heat shock protein (HSP) family genes (HSPA, HSPH, DNAJA, and DNAJB), including some of HSF1-regulated genes, were upregulated under nonstressed conditions in Hikeshi-KO cells (the median of mRNA expression levels in Hikeshi-KO#1 and KO#2 cells relative to that in WT cells is 1.31 and 1.06, respectively) (Fig S7).”

Minor Points

1. Some figure legends in the figure panels are very hard to read ie. the dots that indicate what each sample is are too small and difficult to distinguish. This includes those in Figure 6a and many others. Please make sure all figures are sufficiently reader friendly and easy to distinguish such as those in Figure 6b.

Thank you for the reviewer's suggestion. We changed to the larger dots of the graph legend.

2. In the last paragraph of the results section Figure S6 is cited but should be Figure S7 ?

We did not cite Figure S6 in the last paragraph of the results section, we cited Figure S7.

Reviewer #3 (Comments to the Authors (Required)):

The authors have addressed my previous concerns and I judge this manuscript suitable for publication.

minor text edit:

Figure legend 2D: unclear formulation "normalized to that in each cell transfected with siGL2"

We rewrote the text in Figure legend 2D as follow and changed "siGL2(Ctrl)" in the legend in figure panels to "siGL2(Ctrl)-WT, Hikeshi KO#1, KO#2".

"mRNA expression levels were quantified by qPCR (n=3 biologically independent experiments). In each cell, mRNA expression levels in siHSF1-transfected cells relative to that in siGL2-transfected cells were shown."

April 27, 2022

RE: Life Science Alliance Manuscript #LSA-2021-01241-TRR

Prof. Naoko Imamoto
RIKEN
Cellular Dynamics Laboratory
2-1 Hirosawa
Wako, Saitama 351-0198
Japan

Dear Dr. Imamoto,

Thank you for submitting your Research Article entitled "Lack of Hikeshi activates HSF1 activity under normal conditions and disturbs the heat shock response". It is a pleasure to let you know that your manuscript is now accepted for publication in Life Science Alliance. Congratulations on this interesting work.

DISTRIBUTION OF MATERIALS:

Again, congratulations on a very nice paper. I hope you found the review process to be constructive and are pleased with how the manuscript was handled editorially. We look forward to future exciting submissions from your lab.

Sincerely,
